# APOPT1/COA8 assists COX assembly and is oppositely regulated by UPS and ROS

Alba Signes[1], Raffaele Cerutti[1], Anna S Dickson[2], Cristiane Benincá[1], Elizabeth C Hinchy[1], Daniele Ghezzi[3,4] (iD), Rosalba Carrozzo[5], Enrico Bertini[5] (iD), Michael P Murphy[1], James A Nathan[2] (iD), Carlo Viscomi[1] (iD), Erika Fernandez-Vizarra[1,*] (iD) & Massimo Zeviani[1,**] (iD)

## Abstract

Loss-of-function mutations in *APOPT1*, a gene exclusively found in higher eukaryotes, cause a characteristic type of cavitating leukoencephalopathy associated with mitochondrial cytochrome *c* oxidase (COX) deficiency. Although the genetic association of APOPT1 pathogenic variants with isolated COX defects is now clear, the biochemical link between APOPT1 function and COX has remained elusive. We investigated the molecular role of APOPT1 using different approaches. First, we generated an *Apopt1* knock-out mouse model which shows impaired motor skills, e.g., decreased motor coordination and endurance, associated with reduced COX activity and levels in multiple tissues. In addition, by achieving stable expression of wild-type APOPT1 in control and patient-derived cultured cells we ruled out a role of this protein in apoptosis and established instead that this protein is necessary for proper COX assembly and function. On the other hand, APOPT1 steady-state levels were shown to be controlled by the ubiquitination–proteasome system (UPS). Conversely, in conditions of increased oxidative stress, APOPT1 is stabilized, increasing its mature intramitochondrial form and thereby protecting COX from oxidatively induced degradation.

**Keywords** APOPT1-COA8; cytochrome *c* oxidase; mitochondrial encephalopathy; proteasome–ubiquitin system; reactive oxygen species

**Subject Categories** Genetics, Gene Therapy & Genetic Disease

## Introduction

Isolated mitochondrial cytochrome *c* oxidase (COX) deficiency (OMIM #220110) is one of the most common biochemical presentations in individuals suffering from mitochondrial disease. This enzymatic deficit is associated with diverse clinical presentations, and the phenotypic heterogeneity is mirrored by a large number of mutations reported in more than twenty different genes associated with COX and its assembly (Rak *et al*, 2016).

COX or complex IV (cIV) is the terminal component of the mitochondrial respiratory chain (MRC), catalyzing the oxidation of cytochrome *c* and the reduction of molecular oxygen to water. The energy liberated by this redox reaction sustains the transfer of two protons and four charges across the inner membrane to help sustain the proton electrochemical potential gradient exploited in ATP synthesis, operated by complex V or mitochondrial ATP synthase. Three of the fourteen COX structural subunits, MT-CO1, MT-CO2, and MT-CO3, are encoded in the mitochondrial genome (mtDNA), and mutations in all three have been associated with COX deficiency (Rak *et al*, 2016). The other subunits are nuclear-encoded, and some of them have tissue-specific isoforms, such as COX6A2 and COX7A1 in cardiac and skeletal muscle and COX6B2 in testis, or are expressed in certain physiological conditions such as COX4I2 during hypoxia (Sinkler *et al*, 2017). Mutations in the nuclear-encoded COX structural subunits are much rarer (Massa *et al*, 2008; Shteyer *et al*, 2009; Indrieri *et al*, 2012; Pitceathly *et al*, 2013; Tamiya *et al*, 2014; Hallmann *et al*, 2016; Abu-Libdeh *et al*, 2017; Baertling *et al*, 2017). Nevertheless, a large number of COX deficiency cases arise from mutations in nuclear genes encoding proteins that are involved in cytochrome *c* oxidase biogenesis (Rak *et al*, 2016; Timon-Gomez *et al*, 2018). Information about these COX assembly factors obtained by studying *Saccharomyces cerevisiae* has been very useful in the identification of human orthologs (Tzagoloff & Dieckmann, 1990; Barrientos, 2003; Zee & Glerum, 2006; Soto *et al*, 2012). However, although the absence of these assembly factors leads to impaired formation of fully assembled COX and COX deficiency, their molecular role is still unknown in many cases. Furthermore, it is now clear that there are animal-specific genes encoding COX assembly factors that are not found in yeast (Weraarpachai *et al*, 2009; Martinez Lyons *et al*, 2016; Vidoni *et al*, 2017; Lorenzi *et al*, 2018).

1 MRC-Mitochondrial Biology Unit, University of Cambridge, Cambridge, UK
2 Department of Medicine, Cambridge Institute for Medical Research, University of Cambridge, Cambridge, UK
3 Unit of Medical Genetics and Neurogenetics, Fondazione IRCCS Istituto Neurologico "Carlo Besta", Milan, Italy
4 Department of Pathophysiology and Transplantation, University of Milan, Milan, Italy
5 Unit of Neuromuscular and Neurodegenerative Disorders, Bambino Gesù Children's Research Hospital, IRCCS, Rome, Italy
*Corresponding author. Tel: +44 1223 252846; E-mail: emfvb2@mrc-mbu.cam.ac.uk
**Corresponding author. Tel: +44 1223 252704; E-mail: mdz21@mrc-mbu.cam.ac.uk

Loss-of-function mutations in *APOPT1*, a gene only found in metazoans, have been described as the cause of mitochondrial encephalopathy, characterized by cavitating leukodystrophy with a distinctive MRI pattern, in seven individuals from six different families (Melchionda *et al*, 2014; Sharma *et al*, 2018). Although the genetic association of *APOPT1* pathogenic variants with isolated COX deficiency is well established, the biochemical link between APOPT1 and cIV remains unclear (Melchionda *et al*, 2014). Here, we have characterized the role of APOPT1 in COX maturation in mouse tissues and human cultured cells. We took advantage of the availability of immortalized patient-derived skin fibroblasts (Melchionda *et al*, 2014) and also generated an *Apopt1* knockout mouse model, which displays isolated COX deficiency associated with impaired motor endurance and coordination.

*APOPT1*-less mouse tissues and human fibroblasts showed low steady-state levels of COX subunits and accumulation of early COX assembly intermediates containing the MT-CO1 subunit. In addition, we observed a faster turnover of newly synthesized mtDNA-encoded COX subunits in *APOPT1*-null cells, likely before their incorporation into the COX complex. Oxidative stress enhanced the degradation of these mtDNA-encoded COX subunits, which was prevented by APOPT1. Moreover, the APOPT1 precursor was actively degraded in the cytoplasm by the proteasome, whereas elevated ROS levels stabilized the protein and increased its import into mitochondria. Altogether, these observations indicate a role for APOPT1 in protecting COX assembly from oxidation-induced degradation. While in normal conditions most of the APOPT1 precursor is rapidly destroyed in the cytoplasm by the ubiquitin–proteasome system (UPS), overproduction of mitochondrial ROS stabilizes the protein and increases its mitochondrial import and conversion into the mature, active form.

## Results

### Generation of the *Apopt1* knockout mouse model

CRISPR/Cas9 was used for genomic editing of *Apopt1* to generate a knockout mouse model. To this end, RNA encoding the *Streptococcus pyogenes* Cas9 (SpCas9) plus a customized guide RNA (gRNA) targeting exon 2 were introduced into FVB/NJ 1-day zygotes. The edited embryos were transferred into pseudo-pregnant females and genotyping of the resulting pups identified four founder mice (F0), each carrying several indel modifications. This chimerism could be due to the fact that Cas9 can stay active after several mitotic divisions and that the non-homologous end-joining (NHEJ) DNA repair pathway, which is activated by the CRISPR/Cas 9 editing, is a random process, resulting in multiple mutations. To ensure germline transmission and allow allele segregation, the founder mice were bred with wild-type FVB/NJ mice. Genetic analysis of F1 mice confirmed the presence of heterozygous frameshift indels. After screening of several pups, an individual was selected carrying a substitution of one A with a TG doublet in *Apopt1* exon 2 (c.188delAinsTG, mRNA sequence GenBank NM_026511), which generates a frameshift resulting in a p.Asp55Valfs*20 truncated protein (Fig 1A and B). Crossing of heterozygous mice produced Mendelian ratios of the three possible genotypes, including 25% of mutated homozygous mice

*Apopt1*$^{-/-}$. Apopt1 protein levels could not be analyzed due to the lack of a specific signal for the mouse protein, despite assessing two commercially available and two custom-made antibodies. However, the *Apopt1* mRNA levels skeletal muscle and liver were 25 and 50% in *Apopt1*$^{-/-}$ compared to those found in the corresponding tissues of wild-type *Apopt1*$^{+/+}$ individuals. The amount of *Apopt1* mRNA in the heterozygous mice (*Apopt1*$^{+/-}$) was between those of the +/+ and −/− genotypes (Fig 1C).

### *Apopt1*$^{-/-}$ mice display reduced motor performance

Body weight was not significantly different at 3, 6, or 12 months of age between *Apopt1*$^{-/-}$ and wild-type mice for both genders (Fig 2A), and no major metabolic effects were observed among any of these groups (Appendix Fig S1A–G). *Apopt1*$^{-/-}$ mice did not display feet clasping (Appendix Fig S1J), and the pole-test results, measuring general proprioception, were normal (Appendix Fig S1H). Memory and spatial learning, evaluated by the Y-maze test and measured as percentage of arms alternation, was not affected at any age (Appendix Fig S1I). However, the total number of arm entries was significantly lower in *Apopt1*$^{-/-}$ mice of 6 and 12 months of age (Fig 2D) showing age-related impairment of exploratory behavior. To further explore this, spontaneous movements were monitored in 3- and 12-month-old mice, and a strong reduction of horizontal activity was found only in the older knockout animals, in accordance with the Y-maze test results (Fig 2E). Moreover, both male and female *Apopt1*$^{-/-}$ mice performed significantly worse in the treadmill and rotarod tests at 3 months of age, reflecting impaired motor endurance and coordination, respectively (Fig 2B and C). In order to assess the progression of the phenotype, these tests were repeated with 6- and 12-month-old *Apopt1*$^{-/-}$ mice. However, no further change was observed in their neuromuscular performance as they aged (Fig 2B and C).

### *APOPT1*$^{-/-}$ mice display general COX enzymatic and assembly defects

Skeletal muscle histochemical investigations showed normal fiber morphology in *Apopt1*$^{-/-}$ animals, with no ragged red fibers (Fig 3A). However, COX staining was clearly reduced, whereas succinate dehydrogenase (SDH) staining was normal (Fig 3B). Lower COX staining without any evident histopathological alteration was also observed in the brain and kidney (Fig 3B).

COX enzymatic activity in tissue homogenates confirmed the significantly reduced activity, by 40–60% of controls, in skeletal muscle, kidney, heart, brain, and liver of 3-month-old *Apopt1*$^{-/-}$ mice compared with *Apopt1*$^{+/+}$ or *Apopt1*$^{+/-}$ controls, which showed indistinguishable activities (Fig 3C). One-year-old skeletal muscle, liver and brain still showed significant COX deficits compared with age-matched wild-type animals (Fig 3C). The activities of other respiratory chain complexes and of citrate synthase were the same as controls, except for the liver, where complex I and III activities were also slightly reduced in the *Apopt1*$^{-/-}$ samples (Appendix Fig S2A).

The steady-state levels of COX structural subunits were decreased in *Apopt1*$^{-/-}$ liver, skeletal muscle, and brain (Fig 3D, as well as in cultured MEFs (Appendix Fig S2B). Interestingly, the late (MT-CO3 and COX6B) and intermediate (MT-CO2 and COX5B)

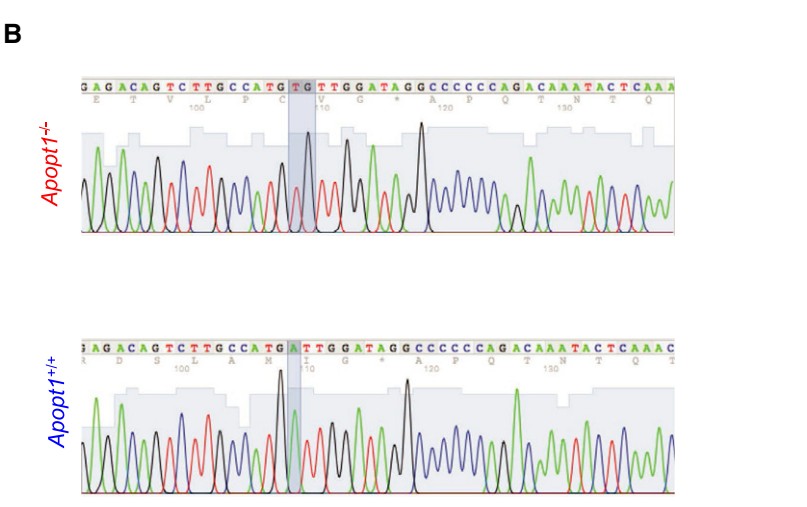

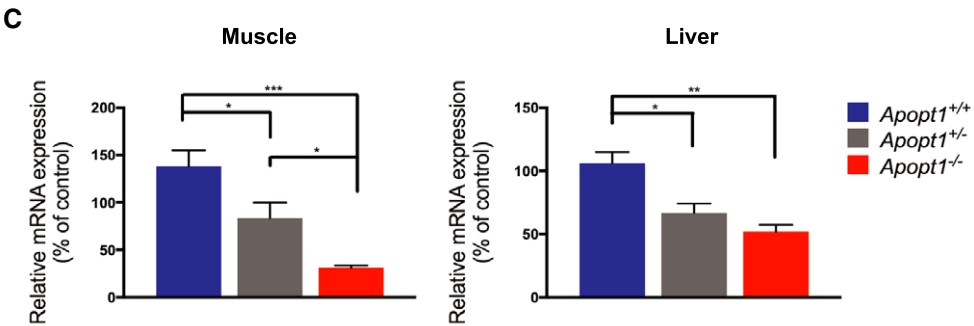

**Figure 1.  Generation of the *Apopt1* knockout mouse model.**

A   Strategy employed for the targeted disruption of mouse chromosome 12 *Apopt1* coding exon 2 by CRISPR/Cas9. The mutation selected for the characterization of the mouse model produced an indel (c.188delAinsTG) resulting in a frameshift and truncated protein (p.Asp55Valfs*20).

B   Chromatograms generated by Sanger sequencing of *Apopt1*⁻/⁻ (KO) and *Apopt1*⁺/⁺ (WT) genomic DNA highlighting the mutated position in comparison with the wild-type sequence.

C   Relative mRNA expression of *Apopt1* normalized to the expression of GAPDH in skeletal muscle and liver of wild-type, heterozygous, and knockout mice of 3 months of age. Data are presented as mean ± SEM ($n$ = 6 mice per genotype). The asterisks represent the significance levels calculated by two-way ANOVA with Sidak's multiple comparisons test: muscle—***$P$ = 0.0001 (WT vs. KO), *$P$ = 0.0310 (WT vs. het), *$P$ = 0.0395 (het vs. KO), liver—**$P$ = 0.0022 (WT vs. KO), *$P$ = 0.0101 (WT vs. het).

assembly subunits as well as MT-CO1 were decreased to a greater extent than the subunits that assemble earlier (COX4 and COX5A; Fig 3D), suggesting that loss of Apopt1 is most detrimental to the intermediate steps of the COX assembly pathway (Vidoni *et al*, 2017).

The amounts of fully assembled complex IV were significantly lower in *Apopt1*$^{-/-}$ skeletal muscle, as shown in one-dimension (1D) blue-native gel electrophoresis (BNGE), Western blot, and immunodetection (Fig 3E) and in cultured MEFs (Appendix Fig S2B). Accumulation of subcomplexes containing the MT-CO1 module (Vidoni *et al*, 2017), also known as "MITRAC" for mito-chondrial translation regulation assembly intermediate of cyto-chrome *c* oxidase (Mick *et al*, 2012), was observed by second dimension (2D) BNGE (Fig 3F). The assembly defect was specific for COX as other respiratory complexes were unaffected.

## APOPT1 is present in a low molecular weight complex in the mitochondrial inner membrane

Human APOPT1 cDNA was amplified from two cell lines: HeLa and HEK 293T. Two different isoforms were detected after cloning and sequencing of the PCR fragments. The first isoform is the transcript containing five coding exons, annotated as APOPT1-201 in Ensembl (www.ensembl.org) with Transcript ID ENST00000409074.6, encoding the full-length protein (Uniprot Q96IL0). The second isoform is APOPT1-203, lacking exon 3 and with Transcript ID ENST00000458117.5, encoding a truncated protein (Uniprot H7C2Z1).

Both isoforms were cloned starting from each of the two putative ATG start codons present in the open reading frame: M1 and M14. These four different cDNAs were fused to GFP in the C terminus and expressed in three different human cell lines (see below). In parallel, a C-terminal HA tag was added to the four cDNA sequences and transfected into 143B osteosarcoma cells. Despite being targeted to mitochondria, the expression levels evaluated by immunofluores-cence of APOPT1-203-M1$^{GFP}$ and APOPT1-203-M14$^{GFP}$ were lower than the full-length APOPT1-201 version in all the tested cell lines (Appendix Fig S3A), and the HA-tagged versions were undetectable by Western blot and immunofluorescence (Appendix Fig S3B and C), which suggests that the protein is not folding correctly and is rapidly degraded. This idea was reinforced by the fact that deletion of *APOPT1* exon 3 causes the pathological phenotype of COX defi-ciency and encephalopathy (Melchionda *et al*, 2014; Sharma *et al*, 2018). Both APOPT1-201-M1 and APOPT1-201-M14 tagged with HA and GFP produce the same size proteins (Appendix Fig S3), being

both targeted to mitochondria, leading to the conclusion that the starting methionine is M14, as was proposed in our original report (Melchionda *et al*, 2014). The tagged APOPT1-201-M1 constructs will be termed APOPT1$^{GFP}$ and APOPT1$^{HA}$, respectively, from now on.

APOPT1$^{GFP}$ protein was detected upon transduction in HeLa, 143B cells, and immortalized control skin fibroblasts, whereas APOPT1$^{HA}$ expression in 143B cells was much lower, indicating that the intrinsic instability of the protein, as previously reported by Melchionda *et al*, was corrected in the GFP chimeric variant. However, contrary to the original report on the identification of APOPT1 (Yasuda *et al*, 2006), we observed no induction of cell death when APOPT1$^{GFP}$ or APOPT1$^{HA}$ was overexpressed (Appendix Fig S4A–C). Therefore, we have no evidence that APOPT1 acts as an apoptogenic factor when overexpressed in human cells. Both APOPT1$^{GFP}$ and APOPT1$^{HA}$ were used to immunodetect the presence of the protein in subcellular fractions. APOPT1 is synthesized as a 22.9-kDa precursor including a mito-chondrial targeting sequence of 26 amino acids which is cleaved off when imported (Melchionda *et al*, 2014), producing a mature protein with a predicted molecular mass of 20.1 kDa. Immortalized fibroblasts derived from patients S2 and S6, carrying homozygous p.Arg79* and heterozygous p.Arg79*/p.Glu124del truncating muta-tions, respectively (Melchionda *et al*, 2014), showed a drastically reduced signal of the band corresponding to the size of mature APOPT1 (20 kDa), when immunodetected with antibodies raised against human APOPT1, which are able to recognize the native protein in human cell lysates (Appendix Fig S5A). The presence or absence of the endogenous precursor protein was impossible to determine as an intense unspecific band of the same electrophoretic mobility as pre-APOPT1 cross-reacts with the anti-APOPT1 antibody (Appendix Fig S5A). On the other hand, the precursor of both tagged versions, APOPT1$^{HA}$ of 24 kDa and APOPT1$^{GFP}$, with a predicted mass of 51 kDa but with an observed migration of 35–40 kDa, were immunovisualized with the specific antibodies against the C-term-inal tags (Fig 4A–C and Appendix Fig S5B).

To determine endogenous and overexpressed tagged APOPT1 localization, we isolated mitochondria from osteosarcoma 143B cells expressing APOPT1$^{HA}$ and APOPT1$^{GFP}$ and from HEK293 human cells. We first tested the association of APOPT1 with the soluble or membrane mitochondrial fractions. Virtually all the endogenous as well as the HA- and GFP-tagged APOPT1 species were associated with the mitochondrial membranes, by a rela-tively loose binding, since carbonate extraction was able to release a large proportion of the protein into the supernatant (Fig 4A and

**Figure 2. Clinical phenotype characterization of the *Apopt1* knockout mice.**

A   Body weight measured in female and male animals at 3, 6, and 12 months of age. Data are presented as mean $\pm$ SEM ($n$ = 5 per group).

B   Distance run by the tested female and male mice on the treadmill at 3 and 12 months of age. Data are presented as mean $\pm$ SEM ($n$ = 6 per group). The asterisks represent the significance levels calculated by two-way ANOVA with Sidak's multiple comparisons test: females—****$P$ < 0.0001 (3 months), ***$P$ = 0.0004 (12 months), males—****$P$ < 0.0001 (3 months), **$P$ = 0.0019 (12 months).

C   Time in seconds spent by the female and male mice on the Rotarod cylinders before falling at different ages. Data are presented as mean $\pm$ SEM ($n$ = 5 per group). The asterisks represent the significance levels calculated by two-way ANOVA with Sidak's multiple comparisons test: females—**$P$ = 0.0059 (3 months), **$P$ = 0.0083 (6 months), *$P$ = 0.0391 (12 months), males—**$P$ = 0.0069 (12 months).

D   Number of entries in each arm of the Y-maze performed at different ages. Data are presented as mean $\pm$ SEM ($n$ = 10 per group). The asterisks represent the significance levels calculated by two-way ANOVA with Sidak's multiple comparisons test: ****$P$ < 0.0001.

E   Total spontaneous horizontal and vertical movements of 12-month-old mice measured in an activity cage. Data are presented as mean $\pm$ SEM ($n$ = 6 per group). The asterisks represent the significance levels calculated by two-way ANOVA with Sidak's multiple comparisons test: ****$P$ < 0.0001 (horizontal).

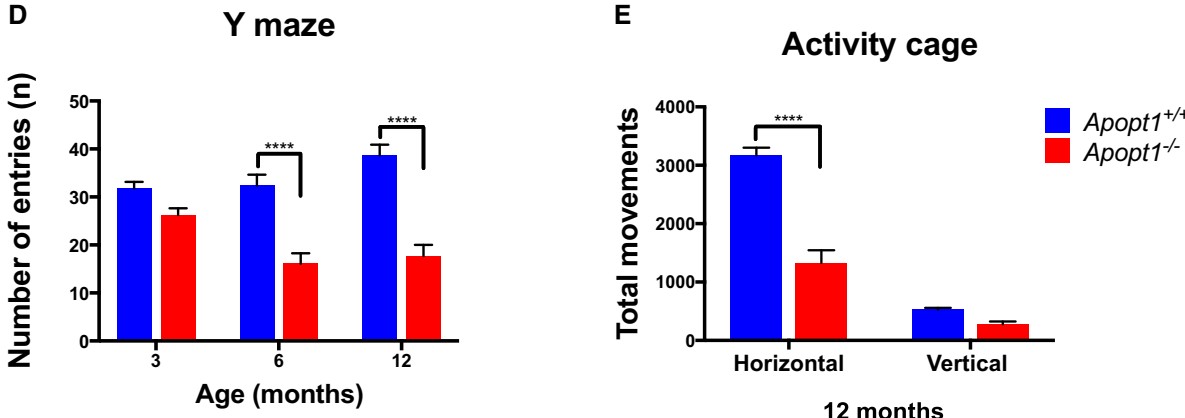

**Figure 2.**

Appendix Fig S5B). Then, we studied the sub-mitochondrial localization in mitoplasts produced by treating mitochondria by either increasing concentrations of digitonin, or hypotonic shock. The mitoplasts were then exposed to trypsin and the integrity and amount of APOPT1 were compared to those of markers specific to the four different mitochondrial compartments: matrix, inner membrane (IM), intermembrane space (IMS), and outer membrane (OM). APOPT1$^{HA}$ showed a trypsin digestion pattern intermediate to AIF or AK2 (IMS markers) and those of COX4 (IM with its N-terminal domain facing the matrix) and ACO2 a soluble matrix protein (Fig 4B and Appendix Fig S5C). This observation was inconclusive to establish whether APOPT1 was facing the matrix or protruding to the IMS. The question was readdressed biochemically by generating mitoplasts using hypotonic shock, incubating the mitochondria in a buffer containing 5 mM sucrose and testing APOPT1$^{HA}$ sensitivity to trypsin compared with different markers. In addition, supernatants were collected after centrifuging the mitochondria in either isotonic or hypotonic buffers, to confirm that hypotonic treatment was associated with the release of IMS proteins. No APOPT1$^{HA}$ was released to the supernatant after hypotonic shock. We then challenged the APOPT1$^{HA}$ found in the pellet with trypsin and found that APOPT1$^{HA}$ sensitivity to the protease was much more similar to that of COX, an IM protein facing the matrix, than that of SCO2, an IM protein facing the IMS (Fig 4C). The difference between the results with digitonin and those with hypotonic shock was likely due to the disruption of IM by high concentrations of digitonin, whereas the hypotonic shock maintained IM integrity. To further validate these results, fluorescence super-resolution microscopy on 143B cells expressing APOPT1$^{GFP}$ showed that GFP fluorescence was contained within the IM, decorated by DS-RED fluorescent protein targeted to IM by the COX8A MTS, and displayed significant co-localization with MitoTracker, staining the IM and the matrix (IM + M), and matrix (M) markers (Fig 4D and Appendix Fig S6). Taken together, these observations indicate that APOPT1 is a protein associated with the inner membrane, with its C terminus located in the matrix.

In addition, Western blot of 2D-BNGE of samples solubilized with the two non-denaturing detergents, n-dodecyl-β-D-maltoside (DDM) and digitonin, revealed that APOPT1$^{HA}$ and APOPT1$^{GFP}$ migrate to the bottom of the gel (Appendix Fig S5D). This suggests that both versions of APOPT1 are not stably interacting in a high molecular weight complex, including COX assembly intermediates containing MT-CO1 or MT-CO2.

## Stable expression of wild-type APOPT1 complements the COX defect in patient-derived fibroblasts

Similar to what was observed in the $Apopt1^{-/-}$ mouse tissues, two skin fibroblast cell lines, derived from two independent patients carrying pathological mutations in $APOPT1$ (S2 and S6), show low steady-state levels of COX subunits, reduced levels of fully assembled COX, and decreased COX activity (Fig 5A–C). Both mutated cell lines were transduced with APOPT1$^{HA}$ and although the protein was always detectable, its amount decreased with time (Fig 5A). Consequently, the complementation of the enzymatic COX defect was partial, especially for S2 (Fig 5A). On the other hand, expression of APOPT1$^{GFP}$ was much more robust and stable, and therefore, transduced S2 and S6 patient cells showed full recovery of COX assembly and COX enzymatic activity, which were around 50% of the control in the non-transduced (naïve) and GFP alone transduced cells (Fig 5B and C).

## APOPT1 participates in the intermediate steps of COX assembly and its absence accelerates the degradation of mtDNA-encoded subunits

The assembly defect in S2 and S6 patient fibroblasts not only leads to an accumulation of subassemblies of the MT-CO1 module (as was also observed in mouse tissues) but it also affects the late COX intermediate containing subunits COX4 and COX5A plus the MT-CO1 and MT-CO2 modules, also known as "S3" (Nijtmans et al, 1998; Vidoni et al, 2017), which is markedly reduced in cells lacking APOPT1 (Fig 5C and D). These COX assembly defects observed in non-transfected or mock-transfected patient-derived fibroblasts were reverted by expression of wild-type APOPT1$^{GFP}$ (Fig 5C and D).

We excluded the possibility that the COX defect in APOPT1-less cells and tissues could be due to a role of APOPT1 in transcription or translation of COX subunits, as no changes were detected in mtDNA-encoded $MT\text{-}CO1$ and $MT\text{-}CO2$ or nuclear-encoded $COX4$ and $COX6B$ mRNA levels (Appendix Fig S7). Furthermore, the synthesis of $^{35}$S labeled mtDNA-encoded proteins was comparable in the mock-transfected patient cells (expressing only GFP) and the controls, i.e., patient cells transfected with APOPT1$^{GFP}$ and wild-type immortalized fibroblasts (Fig 5E). In contrast, when the stability of the protein products was evaluated at different chase times, the amounts of labeled MT-CO2/MT-CO3 in the non-complemented S2 and S6 cells were significantly lower than in the controls after

**Figure 3. Morphological, biochemical, and structural analysis in mouse tissues.**

A  Representative hematoxylin and eosin staining of skeletal muscle in 3-month-old individuals.
B  Histochemical reaction specific to COX and SDH in skeletal muscle, cerebellar cortex, and kidney of the same individuals.
C  COX (CIV) enzymatic activity normalized to the activity of citrate synthase (CS) in five animals per genotype as indicated at 3 and 12 months of age. Data are presented as mean ± SEM ($n$ = 6 per group). The asterisks represent the significance levels calculated by two-way ANOVA with Sidak's multiple comparisons test: Three months: Muscle (SM)—***$P$ = 0.0001 (WT vs. KO), kidney (K)—***$P$ = 0.0002 (WT vs. KO), heart (H)—***$P$ = 0.0001 (WT vs. KO), brain (B) and liver (L)—****$P$ < 0.0001 (WT vs. KO). Twelve months: SM and L—****$P$ < 0.0001, B—***$P$ = 0.0008 (WT vs. KO).
D  Western blot analysis of SDS–PAGE of total lysates from liver, skeletal muscle, and brain from the indicated genotypes, each lane showing the results for one animal. The graph shows the densitometric quantification of the signals obtained in the liver. Data are presented as mean ± SEM ($n$ = 2 WT; $n$ = 2 het; $n$ = 3 KO). The asterisks represent the significance levels calculated by two-way ANOVA with Sidak's multiple comparisons: ***$P$ = 0.0002 and ****$P$ < 0.0001 (WT vs. KO).
E  Western blot analysis of 1D-BNGE of mitochondria from skeletal muscle from the indicated genotypes, each lane showing the results from one animal.
F  Western blot analysis of 2D-BNGE of mitochondria from skeletal muscle from one individual of the indicated genotype. Red arrows indicate the accumulation of COX assembly intermediates in $Apopt1^{-/-}$ samples.

Source data are available online for this figure.

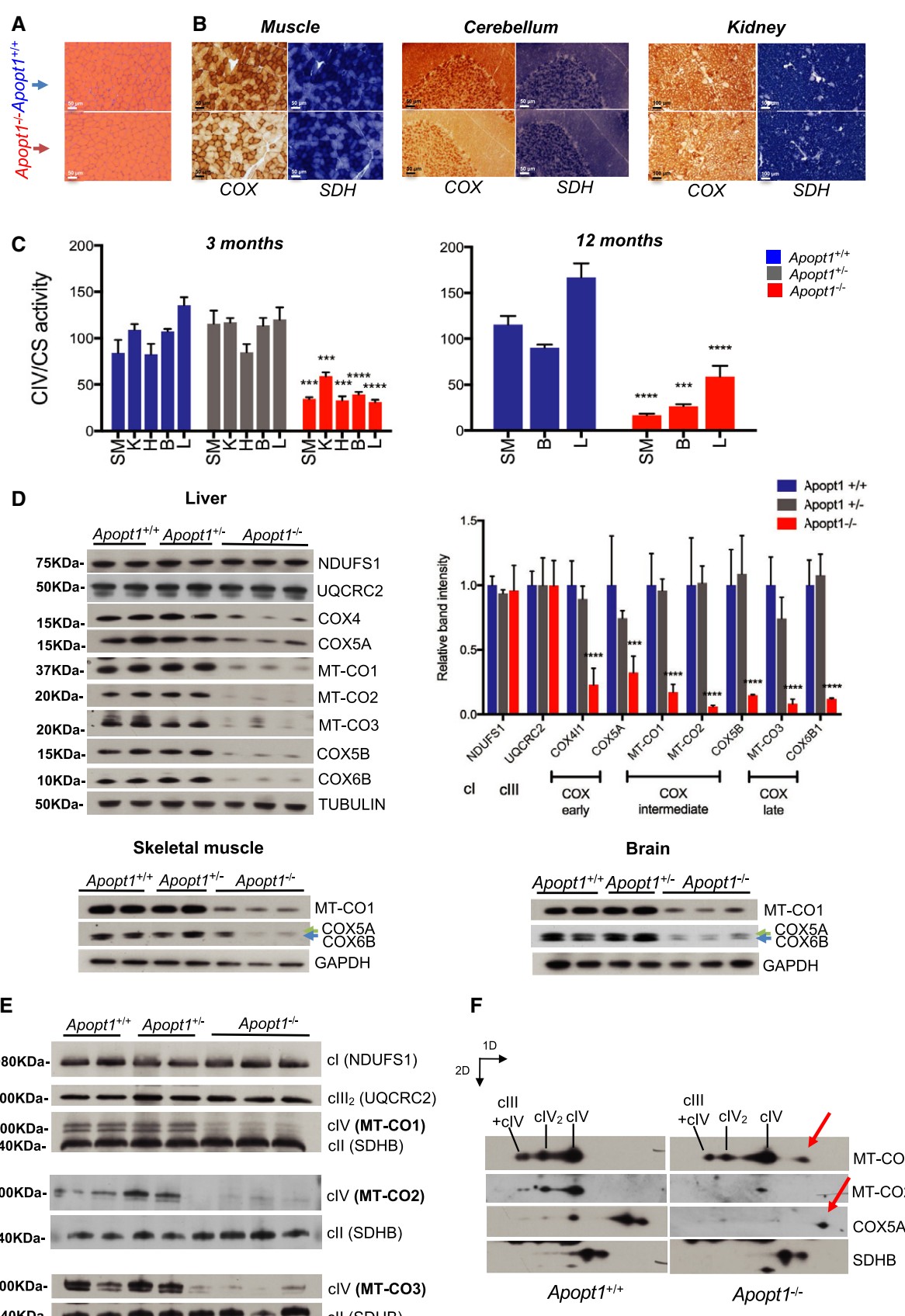

**Figure 3.**

only 3.5 h of culture in "cold" medium, whereas MT-CO1 protein levels were clearly decreased after 6.5 h (Fig 5E). The differences became even more significant at subsequent time points, indicating accelerated degradation of mtDNA-encoded COX subunits in the absence of APOPT1.

### APOPT1 cytoplasmic levels are regulated by the ubiquitination–proteasome system (UPS)

Our original report showed that transfected APOPT1[HA] was stabilized following treatment with the proteasome inhibitor MG132 (Melchionda *et al*, 2014). Further treatments of the newly transduced 143B cells, expressing higher amounts of APOPT1[HA], with MG132 showed an increase in the precursor protein and a corresponding decrease of the mature, cleaved form (Fig 6A). In addition, higher, HA-immunoreactive bands appeared under proteasomal inhibition, which corresponded to ubiquitinated forms of APOPT1, as confirmed by immunoblotting for ubiquitin of immunoprecipitated APOPT1[HA] (Fig 6B).

### Oxidative stress induces APOPT1 stabilization and import into mitochondria where it protects COX from degradation

Our preliminary observations showed that APOPT1 was stabilized when cells were exposed to $H_2O_2$ (Melchionda *et al*, 2014). This phenomenon was confirmed in 143B cells overexpressing APOPT1[HA] and APOPT1[GFP] after a few minutes of exposure to 100 μM $H_2O_2$ (Fig 6C). The rapidity of this response and the consideration that expression of the recombinant protein was under the control of a constitutive exogenous promoter, suggested that this effect was due to a post-translational stabilization of the protein rather than to its increased synthesis. In support of this conclusion, APOPT1 stabilization was not a consequence of proteasome inactivation by oxidative stress (Livnat-Levanon *et al*, 2014; Segref *et al*, 2014) as we did not find an increase in general ubiquitination or in HSP70 levels under exposure to 100 μM $H_2O_2$ (Fig 6C and Appendix Fig S8). Furthermore, pharmacological proteasome inhibition led to the preferential accumulation of the APOPT1 precursor, whereas $H_2O_2$ treatment increased the amounts of the mature species, suggesting either enhanced import or intramitochondrial stabilization, or both. APOPT1 tagged with HA or with GFP

increased 5–10 min after treatment with $H_2O_2$ which continued for up to 6 h, when APOPT1 returned to initial levels (Fig 6C). Other mitochondrial proteins did not vary under these conditions (Fig 6C).

To test whether this effect could be due to ROS selectively generated within mitochondria, we used MitoParaquat (MitoPQ) which is specifically imported into the organelle and generates superoxide ($O_2^{\bullet-}$) only in the mitochondrial matrix (Fig 6C; Robb *et al*, 2015). Superoxide can then be dismutated by MnSOD to $H_2O_2$, which can exit the mitochondria (Brand, 2016; Sies, 2017). Treatment with 5 μM MitoPQ in 143B cells overexpressing APOPT1[HA] and APOPT1[GFP] also promoted the rapid stabilization of both APOPT1 tagged proteins (Fig 6C).

We then treated S6 patient-derived APOPT1-deficient immortalized fibroblasts with 5 μM MitoPQ and analyzed the effect on protein levels for up to 20 h. Similar to what we observed in 143B cells, APOPT1[GFP] levels increased around 4-fold in the complemented fibroblasts after 10 min from the start of MitoPQ treatment, reaching a maximum of 8-fold after 30 min to eventually decrease to pre-treatment levels after 20 h of incubation (Fig 7A and B). The addition of MitoPQ to the culture medium of APOPT1-deficient cells resulted in a gradual decrease of the levels of MT-CO2 and MT-CO1, being the reduction in the latter statistically significant after 20 h of MitoPQ exposure (Fig 7A, C and D). Conversely, wild-type immortalized fibroblasts and APOPT1[GFP] complemented patient cells showed no decrease in the amounts of MT-CO1 and MT-CO2, or even a slight increase, following MitoPQ treatment. The levels of subunit COX5A of complex IV as well as of some subunits of complexes I and III were unaffected by the presence or absence of APOPT1 (Fig 7A).

## Discussion

The results shown here establish a clear link between APOPT1 and COX biogenesis. In our previous report, we tried to validate this association by knocking down *APOPT1* expression by RNAi in different human cells (Melchionda *et al*, 2014). Although *APOPT1* mRNA levels were significantly reduced in the RNAi interfered cell lines, we were unable to demonstrate a clear COX defect, possibly because the residual normal *APOPT1* transcript levels could still be

---

**Figure 4.  Mitochondrial sub-localization of APOPT1.**

A   Western blot of SDS–PAGE of different fractions from 143B cells expressing APOPT1[HA]. Tot: total lysate. Cyto: post-mitochondrial fraction (cytoplasm). Mt: isolated mitochondria. Mt sol: soluble mitochondrial fraction. Mt memb: mitochondrial membranes. $CO_3^{2-}$ pellet: pellet after carbonate extraction with 0.1 M $Na_2CO_3$, pH 10.5 for 30 min. $CO_3^{2-}$ sol: soluble fraction after the carbonate extraction.

B   Western blot of SDS–PAGE of mitochondria used for protease protection assay after digitonin treatment. The experiment was carried out in isolated mitochondria from 143B cells expressing APOPT1[HA] exposed to increasing amounts of digitonin (expressed in μg) and 50 μg/ml trypsin. Complete solubilization with 1% Triton X-100 was used as a control of protease sensitivity. TOM20: Translocase of the outer membrane (OM) 20 kDa. ACO2: aconitase 2 (mitochondrial isoform). AIF: apoptosis-inducing factor. AK2: adenylate kinase 2. OM: outer mitochondrial membrane. IM: inner mitochondrial membrane. IMS: intermembrane space.

C   Western blot of SDS–PAGE of mitochondria used for protease protection assay after hypotonic shock. The experiment was carried out in isolated mitochondria from 143B cells expressing APOPT1[HA] incubated either in isotonic (Iso) or in hypotonic buffers for 5 min (Hypo 5′) or 15 min (Hypo 15′) and 50 μg/ml trypsin. Complete solubilization with 1% Triton X-100 was used as a control of protease sensitivity.

D   N-SIM super-resolution micrographs showing 0.8 μm Maximum Intensity Projection (0.15 μm for each *Z*-stack) of 143B cells expressing APOPT1[GFP] shown in green, specific markers for each mitochondrial compartment: TOMM20 (OM), COX8A (IM), MitoTracker (IM + M), and matrix-target (mScarlet) shown in red. Scale bar: 5 μm. The graph shows Pearson's coefficient in co-localized volume of different sub-compartment combinations with APOPT1[GFP]. Data show mean ± SEM (*n* = 5). One-way ANOVA (**P* = 0.0298 and ***P* = 0.0076) was employed for the statistical analysis.

Source data are available online for this figure.

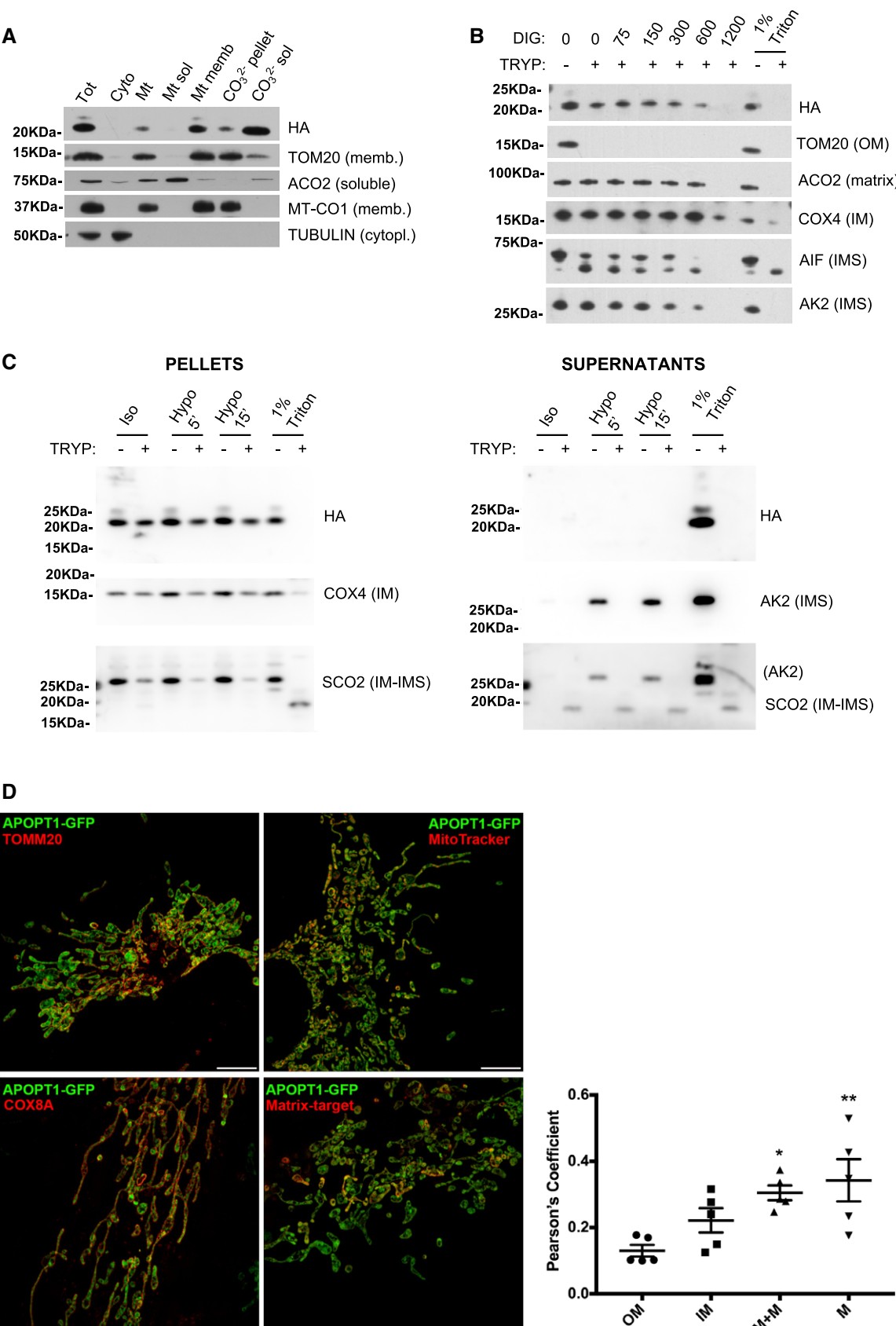

**Figure 4.**

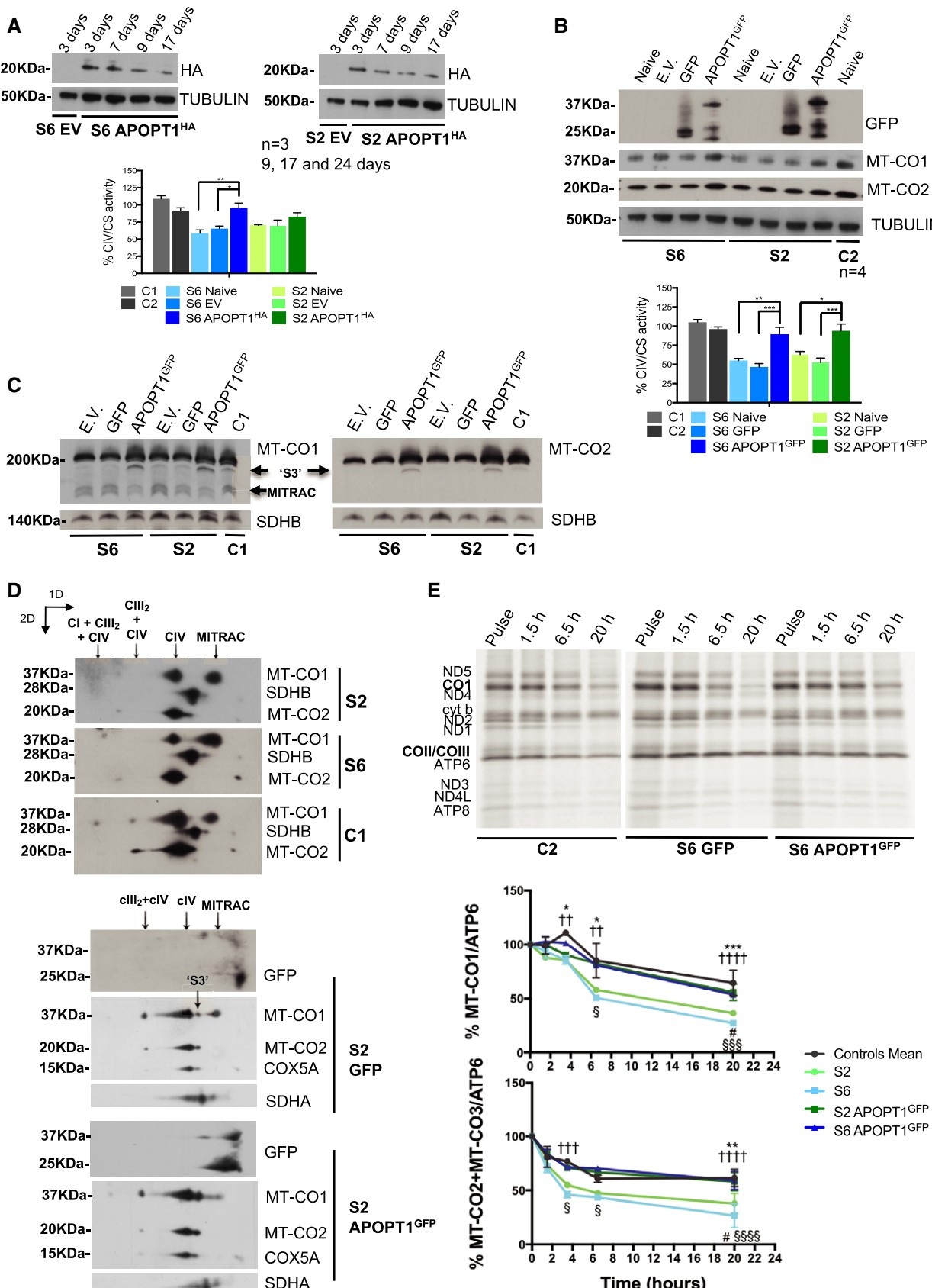

**Figure 5.**

**Figure 5.   Defective COX assembly and accumulation of assembly intermediates in APOPT1-less human cells.**

A   APOPT1[HA] was expressed in S6 and S2 immortalized fibroblasts as shown by the Western blot immunovisualization. The expression levels were tested at different days after transduction. The graph on the right shows the activities of CIV normalized to CS in three biological replicates per cell line. Data are presented as mean ± SEM ($n = 3$). The asterisks represent the significance levels calculated by two-way ANOVA with Tukey's multiple comparisons test: \*\*$P = 0.0030$ (S6 naïve vs. S6 APOPT1[HA]), \*$P = 0.0165$ (S6 EV vs. S6 APOPT1[HA]).

B   APOPT1[GFP] was expressed in S6 and S2 immortalized fibroblasts as shown by the Western blot immunovisualization. The graph on the right shows the activities of CIV normalized to CS in four biological replicates per cell line. Data are presented as mean ± SEM ($n = 4$). The asterisks represent the significance levels calculated by two-way ANOVA with Tukey's multiple comparisons test: \*\*$P = 0.0066$ (S6 naïve vs. S6 APOPT1[GFP]), \*\*\*$P = 0.0006$ (S6 GFP vs. S6 APOPT1[GFP]), \*$P = 0.0169$ (S2 naïve vs. S2 APOPT1[GFP]), \*\*\*$P = 0.0008$ (S2 GFP vs. S2 APOPT1[GFP]).

C   Western blot analysis of 1D-BNGE of mitoplasts extracted from the indicated cell lines. Fully assembled COX migrating at 200 kDa was immunovisualized with antibodies recognizing MT-CO1 and MT-CO2. The presence of the assembly intermediates "MITRAC" and "S3" was also detected (arrows).

D   Western blot analysis of 2D-BNGE of mitoplasts extracted from the indicated cell lines. Fully assembled COX was immunovisualized with antibodies recognizing MT-CO1 and MT-CO2. The presence of the assembly intermediates "MITRAC" and "S3" was also detected (arrows). Anti-GFP immunodetection revealed the presence of the low molecular weight APOPT1[GFP] complex in the complemented cells expressing the tagged protein.

E   L-[35S]-Methionine pulse-chase labeling of mtDNA-encoded proteins. After a 2-h exposure with the radioactive label (pulse), cells were cultured in cold medium for the indicated chase times. The graphs show the densitometric quantification of the bands corresponding to MT-CO1 (left graph) and MT-CO2 + MT-CO3 (right graph) normalized to the ATP6 band over the indicated time points. Graphs represent the values of three biological replicas for each cell line. Data are presented as mean ± SEM ($n = 4$ for controls, $n = 2$ for S2/S6/S2 APOPT1[GFP]/S6 APOPT1[GFP]). The symbols represent the significance levels calculated by two-way ANOVA with Tukey's multiple comparisons test: MT-CO1—3.5 h: \*$P = 0.0202$ (controls vs. S2), ††$P = 0.0039$ (controls vs. S6); 6.5 h: \*$P = 0.0118$ (controls vs. S2), ††$P = 0.0011$ (controls vs. S6), §$P = 0.0163$ (S6 vs. S6 APOPT1[GFP]); 20 h: \*\*\*$P = 0.0002$ (controls vs. S2), ††††$P < 0.0001$ (controls vs. S6), #$P = 0.0343$ (S2 vs. S2 APOPT1[GFP]), §§§$P = 0.0007$ (S6 vs. S6 APOPT1[GFP]). MT-CO2/MT-CO3—3.5 h: †††$P = 0.0003$ (controls vs. S6), §$P = 0.0194$ (S6 vs. S6 APOPT1[GFP]); 6.5 h: §$P = 0.0375$ (S6 vs. S6 APOPT1[GFP]); 20 h: \*\*$P = 0.0013$ (controls vs. S2), ††††$P < 0.0001$ (controls vs. S6), #$P = 0.0234$ (S2 vs. S2 APOPT1[GFP]), §§§§$P < 0.0001$ (S6 vs. S6 APOPT1[GFP]).

Data information: C1 and C2: control immortalized fibroblasts. EV: Cells transduced with the empty vector. Naïve: Non-transduced immortalized fibroblasts. GFP: Cells transduced with a recombinant GFP expression vector. Tubulin, SDHB, and SDHA were used as loading controls in the different experiments as indicated.
Source data are available online for this figure.

translated to give a sufficient amount of protein to achieve normal COX assembly. In support of this possibility stands, the observation that all the mutations so far reported in APOPT1 patients carried either homozygous or compound heterozygous loss-of-function mutations which prevented the synthesis of a full-length protein. In addition, acute shRNA treatment in control immortalized fibroblasts induced cell death (Melchionda *et al*, 2014), a phenomenon for which we still have no explanation since APOPT1-null patient fibroblasts show normal growth in standard culture conditions. Contrariwise, APOPT1 was originally described as an inducer of cell death by apoptosis (Yasuda *et al*, 2006; Sun *et al*, 2008). However, we did not observe any cell death or growth arrest when overexpressing APOPT1 in three different human cell lines, including HeLa transduced with APOPT1 tagged with a C-terminal GFP, as in the original report (Yasuda *et al*, 2006).

The genetic knockout mouse model of *Apopt1* showed global COX deficiency with reduced enzymatic activity, low steady-state levels of structural subunits and defective assembly in all the tested tissues. The *in vivo* mouse model was also exploited to evaluate the impact of *Apopt1* deletion on metabolic, neurological, and motor phenotypes. *Apopt1*$^{-/-}$ mice show significantly impaired motor coordination and endurance, indicative of neurological and muscular involvement in the clinical phenotype, thus confirming *APOPT1* as a disease gene. In humans, APOPT1 deficiency is associated with a very characteristic cavitating leukoencephalopathy, which may prompt to specific analysis of the *APOPT1* gene (Sharma *et al*, 2018).

The transfection and transduction systems previously used did not allow the stable expression of the HA-tagged APOPT1 protein in human fibroblasts, despite being highly overexpressed at the mRNA level, thus preventing the functional characterization of APOPT1 (Melchionda *et al*, 2014). By optimizing the transduction and expression systems with different viral vectors and recombinant constructs, we have achieved stable translation of wild-type human APOPT1[HA] and APOPT1[GFP] in different human tumor cell lines and immortalized fibroblasts. This fact, plus the development of polyclonal antibodies recognizing the endogenous human protein, enabled us to reliably determine APOPT1 mitochondrial and

**Figure 6.   Effects of proteasome inhibition and ROS overproduction on APOPT1 stability.**

A   Western blot analysis of SDS–PAGE of MGM132-treated 143B APOPT1[HA] cells. The graph represents the densitometric quantification of the signals for the precursor and mature protein.

B   The upper part of the panel (Input) shows Western blot analyses of APOPT1[HA] in the MGM132-treated cells. Note the appearance of higher molecular weight bands upon longer exposure in the samples treated with the proteasome inhibitor. The bottom part of the panel (Purified fractions) shows the analysis of fractions from the same cells immunoprecipitated with anti-HA. Note that the higher molecular weight species are cross-reacting with both anti-HA and anti-ubiquitin.

C   Western blot analysis of SDS–PAGE of total lysates from 143B cells overexpressing tagged APOPT1 (as indicated) and exposed to 100 μM H₂O₂, as illustrated by the scheme (H₂O₂ treatment), for the indicated times. The graphs represent the densitometric quantification of the tagged APOPT1 signal at each time point. The graph inset shows that the increase of APOPT1 occurs in the first minutes after the exposure to H₂O₂.

D   Western blot analysis of SDS–PAGE of total lysates from 143B cells overexpressing tagged APOPT1 (as indicated) and exposed to 5 μM MitoParaquat (MitoPQ), as illustrated by the scheme (MitoPQ treatment), for the indicated times. The graphs represent the densitometric quantification of the tagged APOPT1 signal at each time point. The graph inset shows that the increase of APOPT1 occurs in the first minutes after the exposure to MitoPQ.

Source data are available online for this figure.

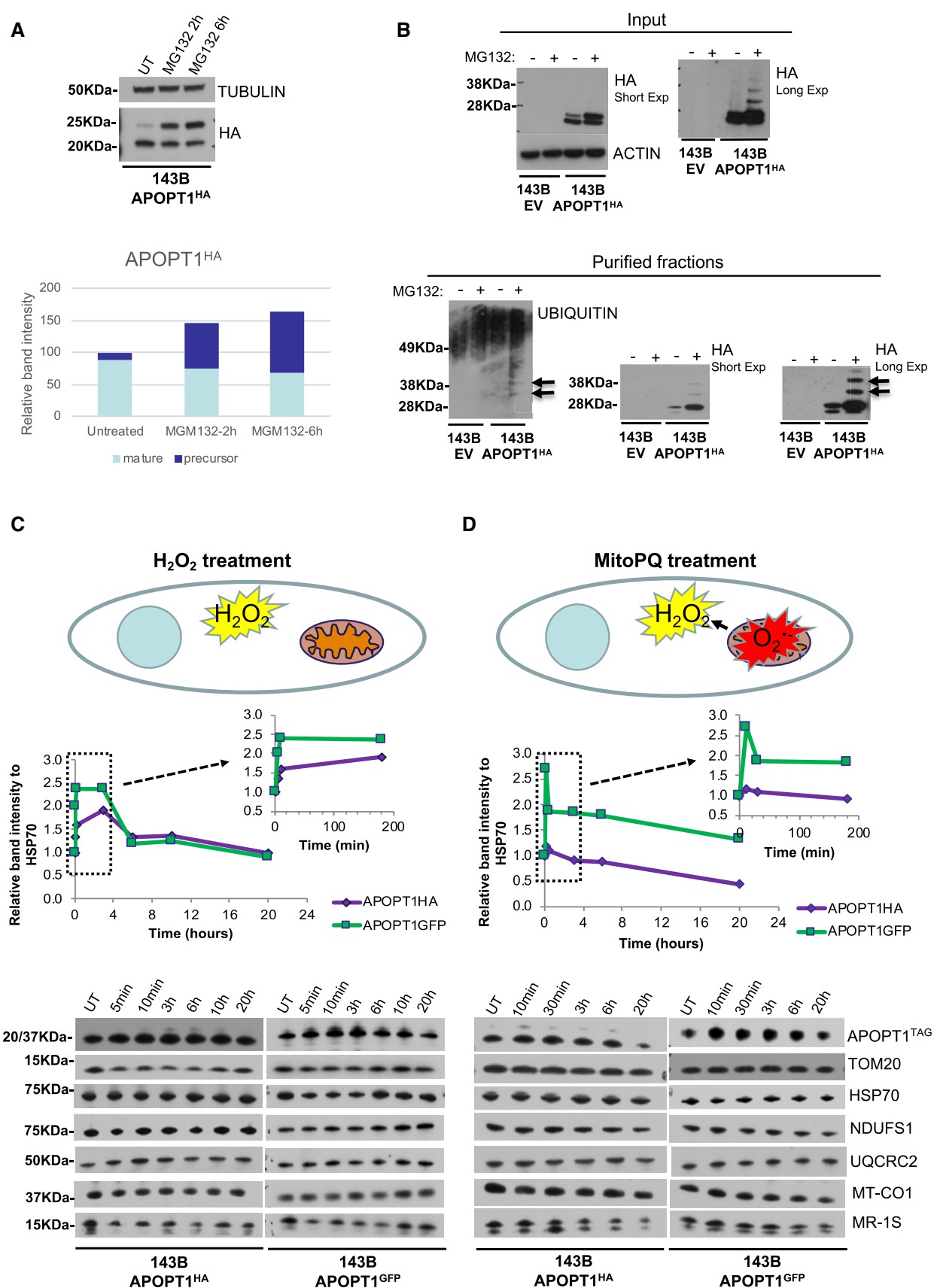

**Figure 6.**

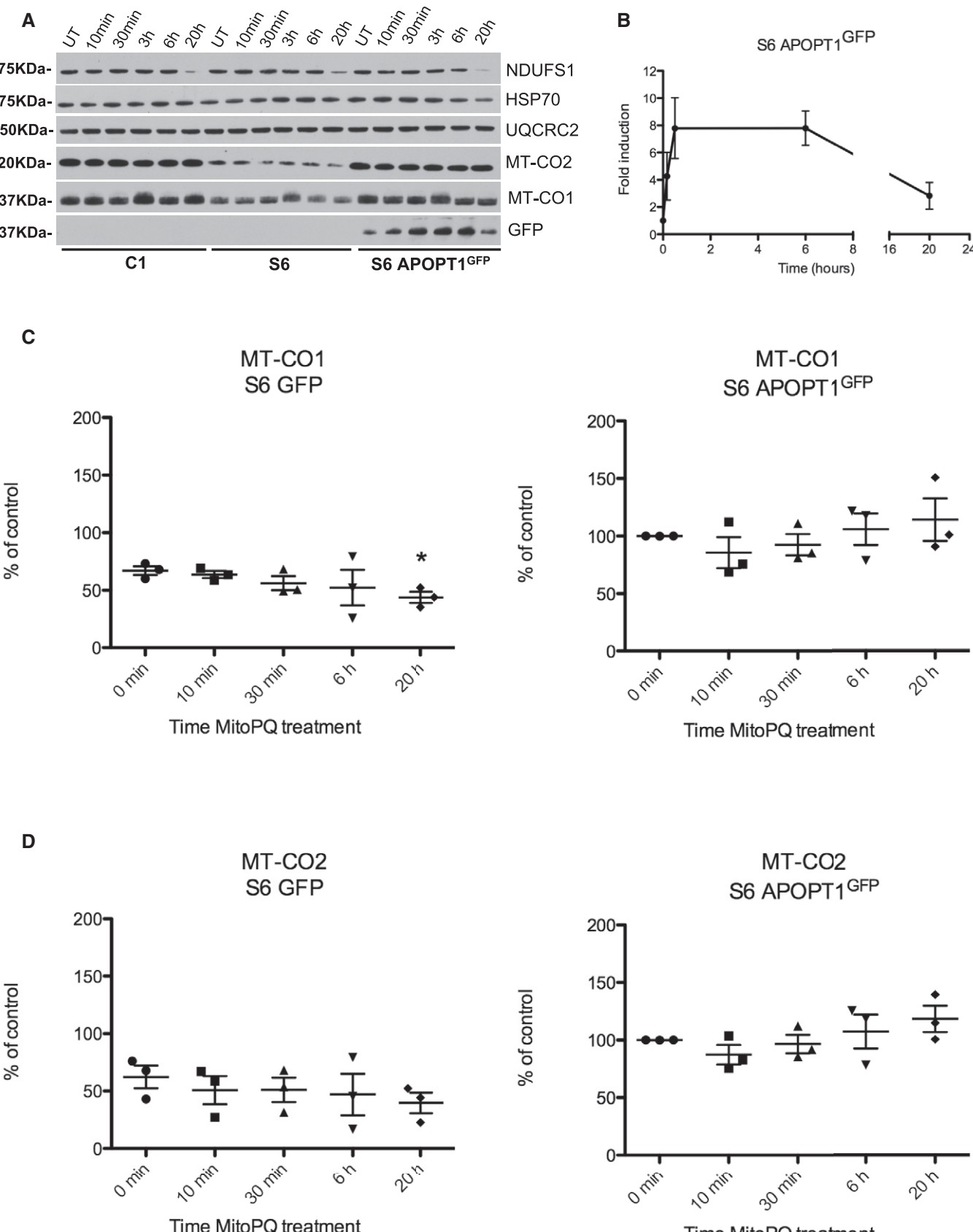

Figure 7.

**Figure 7. Effects of MitoPQ treatment in the absence or presence of APOPT1.**

A   Western blot analysis of SDS–PAGE of different mitochondrial proteins in total lysates from the indicated cell lines treated with 5 μM MitoPQ at the indicated times. UT: untreated cells.

B   Densitometric quantification of APOPT1$^{GFP}$ signal during the treatment in two biological replicas. Data are presented as mean $\pm$ SEM ($n$ = 2).

C   Densitometric quantification of MT-CO1 signal in the non-complemented APOPT1-less cells (S6 GFP) vs. the complemented cells (S6 APOPT1$^{GFP}$). Three biological replicas and two technical replicates were carried out for each cell line. The signals in UT S6 APOPT1 were considered 100%. Data are presented as mean $\pm$ SEM ($n$ = 3). The levels of MT-CO1 were significantly lower in the S6 GFP patient cells after 20 h of MitoPQ treatment compared to the untreated cells (*$P$ = 0.0202, two-tailed unpaired Student's $t$-test).

D   Densitometric quantification of MT-CO2 signal in the non-complemented APOPT1-less cells (S6 GFP) vs. the complemented cells (S6 APOPT1$^{GFP}$). Three biological replicas and two technical replicates were carried out for each cell line. The signals in UT S6 APOPT1 were considered 100%. Data are presented as mean $\pm$ SEM ($n$ = 3).

Source data are available online for this figure.

sub-mitochondrial localization as a protein associated to the inner membrane and facing the IMS. In addition, transduction of APOPT1$^{HA}$ and APOPT1$^{GFP}$ in immortalized fibroblasts derived from two unrelated patients, S2 and S6 (Melchionda *et al*, 2014), rescued the enzymatic and assembly COX defect. Taken together, the evidence collected from the *in vivo* and cell models establishes unequivocally the involvement of APOPT1 in COX biogenesis and excludes a role of this protein in apoptosis, at least in the cells and tissues considered in our study. Therefore, we propose that its name be changed to cytochrome *c* oxidase assembly factor (COA) 8, adding to the collection of already known COX assembly factors named COA 1–7.

The assembly defect shown by all the APOPT1-null models analyzed, i.e., *Apopt1*$^{-/-}$ tissues as well as patient-derived immortalized fibroblasts, involves the global down-regulation of COX with an accumulation of subcomplexes including early assembly subunits (COX4 and COX5A) and the MT-CO1 module (Vidoni *et al*, 2017; largely coincidental with the MITRAC complex (Mick *et al*, 2012)). Consistent with this observation, COX4, COX5A, and MITRAC steady-state levels were less reduced than those of the MT-CO2 and MT-CO3 modules, involved in later steps of COX assembly (Vidoni *et al*, 2017; Timon-Gomez *et al*, 2018). Moreover, the "S3" subassembly, containing the MT-CO1 and MT-CO2 modules together, is markedly reduced in the patient-derived cells and mouse tissues. Thus, APOPT1/COA8 must play a role in joining or stabilizing the MT-CO2 module to COX4-COX5A and MT-CO1. Absence of APOPT1/COA8 does not affect the synthesis of any of the mtDNA-encoded COX subunits. However, their stability is severely compromised, being most probably actively degraded due to impaired incorporation into the nascent complex. This same phenomenon has been consistently observed when different COX assembly factors, such as SCO1, COX20, CMC1, or COX18, are mutated or absent in human cells, determining the stalling in the assembly of the MT-CO1 module (Leary *et al*, 2009; Bourens *et al*, 2014; Bourens & Barrientos, 2017a, b). Accordingly, it has been shown that COX-deficient *S. cerevisiae* strains showing high sensitivity to hydrogen peroxide and an accumulation of subassemblies containing hemylated Cox1 (Khalimonchuk *et al*, 2007, Veniamin *et al*, 2011) display a faster turnover of unassembled COX subunits, which is mediated by the ATPase Afg1 (Khalimonchuk *et al*, 2007). Interestingly, LACE1, the human orthologue of Afg1, has also been shown to be involved in the degradation of nuclear-encoded COX subunits (Cesnekova *et al*, 2016). Therefore, these lines of evidence strongly suggest that there is a regulatory mechanism of COX assembly that links the accumulation of MT-CO1 containing subassemblies with faster degradation of unassembled COX subunits. In the case of APOPT1 deficiency,

neither the human cultured cells nor the mouse tissues showed increased H$_2$O$_2$ production in non-induced conditions (results not shown); however, when the patient-derived fibroblasts were oxidatively challenged, they showed a significant increase in ROS production compared to the controls (Melchionda *et al*, 2014), which argues in favor of a "pro-oxidant state" in the absence of APOPT1. In addition, these same APOPT1-null cells showed further reduction in COX levels when oxidants were added to the culture medium. The presence of APOPT1 in control cells (both complemented patient and control fibroblasts) protected COX from this damage. These observations underscore the association of APOPT1 function with COX assembly and stabilization of the enzyme as well as with the protection of the nascent enzyme from oxidative damage, which leads to degradation of its structural components.

Concerning the regulation of APOPT1/COA8, the results presented here indicated active ubiquitination and proteasome-mediated degradation of the APOPT1 precursor in the cytoplasm. UPS-mediated degradation of mitochondria-targeted proteins has been proposed as a regulatory mechanism to prevent the accumulation of these precursor proteins in the cytoplasm and consequent cellular stress (Bragoszewski *et al*, 2013; Wrobel *et al*, 2015). In addition, the UPS is involved in several other processes controlling mitochondrial fitness, such as fission and fusion, PINK1 and PARKIN-mediated mitophagy and regulation of the OM and IMS mitochondrial proteome (reviewed in Bragoszewski *et al*, 2017). However, this kind of mechanism has never been described in association with COX biogenesis. In addition, in short-term and mild oxidative stress conditions, APOPT1/COA8 import inside mitochondria is enhanced. This phenomenon cannot be attributed to direct proteasome inhibition by H$_2$O$_2$ (Livnat-Levanon *et al*, 2014; Segref *et al*, 2014) as we did not observe signs of accumulation of ubiquitinated proteins or increased heat-shock protein response. Moreover, pharmacological inhibition of the proteasome produced the preferential accumulation of APOPT1 precursor and not of the mature protein, indicating that a significant proportion of newly synthesized APOPT1 is normally ubiquitinated and degraded by the proteasome in the cytoplasm before mitochondrial import. On the other hand, oxidative stress induced by direct addition of H$_2$O$_2$ in the cell culture medium or via MitoPQ increased the amount of the mature protein, suggesting that H$_2$O$_2$ outside mitochondria and/or ROS produced within the organelle could be the signal to activate or unblock APOPT1/COA8 import, as well and to stabilize it inside mitochondria. It is well known that H$_2$O$_2$ can generate regulatory signals by oxidative modification of key Cys residues in responsive proteins (Hurd *et al*, 2005; Murphy, 2009; Hamanaka & Chandel, 2010; Brand, 2016; van Leeuwen *et al*, 2017; Sies, 2017). Human and

mouse APOPT1 have Cys residues eleven amino acids upstream and nineteen amino acids downstream of the predicted MTS cleavage site, which are conserved in mammalian species. In fact, oxidative folding mediated by Cys residues is a very well-studied mechanism in yeast, necessary for the import and regulation of IMS proteins through the MIA (mitochondrial IMS import and assembly) pathway (Wasilewski *et al*, 2017), including some COX structural subunits and assembly factors (Khalimonchuk & Winge, 2008). Oxidative regulation of the import of mitochondrial inner membrane or matrix proteins has not been described before and further experimental demonstration will be needed to test the possibility of a $H_2O_2$-mediated APOPT1/COA8 structural change through oxidative modification of its Cys residues to favor its import inside mitochondria.

In summary, here we have conclusively demonstrated that genetic ablation of *APOPT1* is directly related to COX deficiency and mitochondrial disease. Moreover, we propose a mechanism of modulation of COX assembly that is mediated by regulating the levels of APOPT1/COA8, first in the cytoplasm by degrading it through the UPS and secondly by ROS, which stimulates its import into mitochondria to promote COX assembly at intermediate steps by stabilizing and/or protecting the COX subassemblies from oxidative damage. Further experimental work will be undertaken to understand the specific ROS-mediated molecular mechanism by which APOPT1 is imported inside mitochondria and what proteins bind and collaborate with it.

# Materials and Methods

### Generation of *APOPT1*$^{-/-}$ mice

The gRNA is composed of a "scaffold" sequence, necessary for the binding of the Cas, and a user-defined 20-nucleotide sequence, known as "spacer", that determines the genomic region to be targeted. The "spacer" was designed using the next web tool: http://crispr.mit.edu/ and was targeted to the exon 2 of the mouse *Apopt1* gene (GenBank ID: 68020). The sequence with the highest quality score, meaning perfect homology with the target with an NGG sequence at the 3′ end (PAM sequence, necessary for the binding and activation of the Cas9) and no off-targets in any other gene, was: 5′-CTGGGGGGGCCTATCCAATCATGG-3′. The plasmid pSpCas9 (BB)-PX330 (Addgene plasmid # 42230) was used as a template to amplify by PCR the gRNA construct ("spacer" + "scaffold") preceded by a T3 promoter to allow subsequent *in vitro* transcription. The product amplified was then cloned into the pCR2.1 TOPO TA cloning kit (Invitrogen), and the resulting plasmid was linearized by digestion with the restriction enzyme EcoRI and transcribed *in vitro* (Riboprobe system-T3, Promega). The commercial plasmid encoding the SpCas9 (Addgene plasmid # 48625) was also linearized by digestion with SphI and transcribed *in vitro* (Riboprobe system-T3, Promega). After RNA purification of the gRNA and Cas9, the sequences were sent for microinjection into fertilized FVB/NJ one-cell embryos (Core Facility for Conditional Mutagenesis, Milan).

### Animal work

All procedures were conducted under the 1986 UK Animals (Scientific Procedures) Act and approved by Home Office license (PPL: 7538 and P6C97520A) and local ethical review. The mice were kept on FVB/NJ background, and wild-type littermates were used as controls. The animals were maintained in a controlled environment within the animal care facility in the Phenomics Laboratory, Forvie Site, Cambridge Biomedical Campus, Cambridge, UK. Animals of both gender at 3, 6, and 12 months of age, as indicated in the legends of the figures, were used for the various phenotype analyses.

### Histopathological analysis

Mice were euthanized by cervical dislocation, and organs were quickly dissected, frozen by immersing in isopentane cooled with liquid nitrogen, and stored in cryovials in a −80°C freezer. Tissues were then sectioned in a cryostat, placed in slides. The slides were then stained for COX and SDH, as described (Sciacco & Bonilla, 1996). For histological and immunohistochemical analyses, mice were anesthetized with an overdose of pentobarbital and perfused with PBS followed by 10% neutral buffer formalin (NBF). Hematoxylin–eosin and Nissl staining were performed by standard methods.

### Behavioral tests

Mice were monitored weekly to examine body condition, weight loss, and general health. Neurological, motor, and metabolic phenotype was further evaluated with the set of tests described below.

#### Hindlimb clasping
Mice were grasped from the base of the tail, lifted clear of all surrounding objects and their hindlimb position observed for 10 s. Normal position was defined as hindlimbs splayed outward, away from the abdomen, and abnormal as one or both hindlimbs retracted toward the abdomen.

#### Comprehensive laboratory animal monitoring system (CLAMS)
*Apopt1*$^{-/-}$ mice and control littermates were individually placed in the CLAMS™ system of metabolic cages and monitored over a 36-h period. Data were collected every 10-min. The parameters analyzed were: $VO_2$ (volume of oxygen consumed, ml/kg/h), $VCO_2$ (volume of carbon dioxide produced, ml/kg/h), RER (respiratory exchange ratio), heat (kcal/h), XYZ total activity, ambulatory movements, and rear movements.

#### Treadmill
Mice were forced to run to exhaustion (defined as > 10 falls/min) in a treadmill apparatus (Columbus Instruments, Columbus, OH). One trial for two consecutive days was conducted prior to testing to allow the mice enough time to acclimatize. On the test day, the treadmill was set to an angle of inclination of 10°. The speed was initially set at 11 m/min for 3 min, and it was increased 0.3 m/min every 3 min up to a maximum speed of 75 m/min.

#### Rotarod
A rotarod apparatus (Ugo Basile, Italy) was used to assess motor performance and coordination. One trial for two consecutive days was conducted prior to testing to allow the mice enough time to acclimatize. On the test day, three trials were completed setting the rod to accelerate from 2 to 40 rpm in 300 s. The latency to fall was

recorded. Mice were returned to their home cage during the inter-trial interval of 15–20 min.

### Y-maze spontaneous alternation

The test was conducted in a large Y-shaped maze with three equal arms of 40 cm length, 8 cm width, and 15 cm height, attached at 120° angle from each other. Mice were placed at the center of the maze and allowed to freely explore the three arms for 5 min. No acclimatization was required as this test evaluates the willingness of mice to explore new environments. The sequence of arm entries was manually recorded and then the total number of entries plotted and the percentage of alternation calculated as:

$$\% \text{ alternation} = \left( \frac{\text{actual alternation}}{\text{maximum alternation}} \right) \times 100$$

Actual alternation = number of correct alternations (defined as entry into all three arms consecutively); Maximum alternation = total number of entries minus two

### Pole test

Mice were placed head-upward on the top of a vertical rough-surfaced pole (diameter 5 mm; height 50 cm) and the time to orientate downward and descend it was recorded with a maximum duration of 60 s. Three trials for two consecutive days were conducted prior to testing to allow the mice enough time to acclimatize. On the test day, animals received three trials and the time to descend the pole was recorded.

### Activity cage

Mice were individually placed in the center of an activity cage (Ugo Basile, Italy) and left 5 min to freely explore before commencing the test to allow them enough time to acclimatize. After that, horizontal and vertical movements were recorded at intervals of 1 min for 30 min.

### Mitochondrial extraction and subfractionation

Mitochondrial isolation from mouse tissues and cultured cells was performed as described (Fernandez-Vizarra et al, 2006; Fernández-Vizarra et al, 2010).

In order to separate the soluble and membrane fractions, freshly isolated mitochondria were sonicated three times and then centrifuged at $100,000 \times g$ for 30 min at 4°C to separate the supernatant containing the soluble proteins and the membrane-associated proteins in pellet (Ghezzi et al, 2009). To split the peripherally bound from the integral membrane proteins, the pellets obtained in the previous centrifugation step were resuspended in a buffer containing 0.1 M $Na_2CO_3$, pH 10.5, 0.25 M sucrose and 0.2 mM EDTA; incubated for 30 min on ice and then centrifuged at $100,000 \times g$ for 30 min at 4°C, to separate the pellet the from the supernatant containing the loosely bound membrane proteins (Satoh et al, 2003).

For sub-mitochondrial localization, mitochondria were isolated as described above, split in 0.5 mg aliquots and treated with increasing digitonin amounts (from 75 to 1,200 µg) and 50 µg/ml trypsin at room temperature for 30 min. To assess mitochondria integrity, one sample was treated with protease but no digitonin.

As a control of protein digestion, one sample was treated with trypsin in the presence of 1% Triton X-100. For the hypotonic shock, mitochondria were incubated in either isotonic buffer (320 mM sucrose, 10 mM Tris–HCl pH 7.4, 1 mM EDTA), hypotonic buffer (5 mM sucrose, 10 mM Tris–HCl pH 7.4, 1 mM EDTA), for either 5 or 15 min, or completely solubilized with 1% Triton X-100. One-half of each sample was then treated with 50 µg/ml trypsin at room temperature for 30 min. Afterward, mitochondria from trypsin-treated and untreated samples were centrifuged at $9,000 \times g$ for 10 min at 4°C, and the supernatant was separated from the pellets. The pellets were washed twice with isotonic buffer and finally solubilized with 1% SDS for SDS–PAGE and WB analysis.

### Enzymatic assays

The activities of the respiratory chain enzymes and citrate synthase (CS) were measured in mitochondria-enriched tissue samples and in digitonin solubilized cell samples by spectrophotometry in a 96-well plate reader as described previously (Tiranti et al, 1995; Kirby et al, 2007), with slight modifications.

### Human samples

The primary and immortalized cultured fibroblasts were derived from patients S2 and S6, reported in Melchionda et al (2014) and were provided by the cellular BioBank of Telethon Italy located at the Fondazione Istituto Neurologico Carlo Besta, Milan, Italy. Informed consent was obtained at the time of biopsy collection under local ethical committee approval. All the procedures involving human samples were performed under the principles set out in the WMA Declaration of Helsinki and the Department of Health and Human Services Belmont Report.

### Cell culture

Human HeLa, 143B osteosarcoma cells, and primary and immortalized skin fibroblasts were grown in high-glucose DMEM plus Glutamax® and sodium pyruvate supplemented with 10% fetal bovine serum (FBS), 1× penicillin and streptomycin (all from Gibco-Life Technologies), and 50 µg/ml uridine (Sigma) in humidified atmosphere at 37°C and 5% $CO_2$. Selective media were prepared adding 1 µg/ml puromycin or 100 µg/ml hygromycin (both from Invivo-Gen).

### Oxidative stress in cell cultures

143B cells overexpressing APOPT1$^{HA}$ or APOPT1$^{GFP}$ were treated with 100 µM $H_2O_2$ for 5 and 10 min and for 3, 6, 10, and 20 h. Same cell lines plus wild-type, mutated, and APOPT1$^{GFP}$ complemented fibroblasts were treated with 5 µM MitoParaquat for 10 and 30 min and for 3, 6, and 20 h. Samples were then lysed in RIPA buffer (50 mM Tris pH 7.4, 0.1% SDS, 1% NP-40, 0.5% Na deoxycholate, 150 mM NaCl) with the addition of protease inhibitors (cOmplete™ Mini EDTA-free Protease Inhibitor Cocktail, 100 mM NEM and 100 mM IAA). Lysates were centrifuged at $16,900 \times g$ for 20 min; supernatants were mixed with loading buffer and separated by SDS–PAGE prior to immunoblotting.

## Proteasome inhibition and ubiquitination assay

143B empty vector (EV), APOPT1[HA] cells were treated with 10 μM MG132 for 2 or 6 h. For the immunoprecipitation of APOPT1[HA] to assess for ubiquitination, $1 \times 10^7$ 143B EV or 143B APOPT1[HA] cells were treated with or without MG132 for 2 h and lysed in RIPA buffer (50 mM Tris pH 7.4, 0.1% SDS, 1% NP-40, 0.5% Na deoxycholate, 150 mM NaCl) with the addition of protease inhibitors (cOmplete™ Mini EDTA-free Protease Inhibitor Cocktail, 100 mM NEM, and 100 mM IAA). Lysates were centrifuged at $16,900 \times g$ for 10 min. The supernatant was centrifuged at $135,800 \times g$ for 1 h, before samples were pre-cleared using sepharose CL4B for (1 h, 4°C). Samples were then incubated with 10 μl EZviewTM Red Anti-HA beads (Sigma-Aldrich) overnight at 4°C. Resins were washed five times with RIPA buffer, and the bound proteins were eluted using 40 μl 100 μg/ml HA peptide (Sigma-Aldrich; in 0.5% NP-40 with protease inhibitors) for 1 h at 4°C. Protein samples in loading buffer were heated at 75°C for 10 min and separated by SDS–PAGE prior to immunoblotting.

## Cell growth

Growth curves were assessed using an Incucyte ZOOM or an Incucyte HD instrument (Essen Bioscience) using an algorithm to calculate cell confluency based on microscope imaging of the plates. Images were taken every 2 h for a total period of 4 days.

## Cloning of APOPT1 cDNA and lentiviral transduction

For the amplification of APOPT1 cDNA, total RNA was extracted from HeLa and HEK293T cells using the TRIzol Plus RNA Purification System (Invitrogen-ThermoFisher Scientific) and retrotranscribed with the Omniscript RT kit (Qiagen). Approximately 200 ng of cDNA was used as template for the amplification of APOPT1 using specific primers (Table 1). C-terminal HA tags were added by PCR amplification, and the GFP tag was added by cloning a stop codon-less APOPT1 cDNA in frame with EGFP already inserted into pCR2.1. The PCR generated fragments in pCR2.1 TA cloning vector (Invitrogen) were then cloned into pWPXLd-ires-Puro[R] and pWPXLd-ires-Hygro[R] lentiviral expression vectors, modified versions of pWPXLd (Addgene #12258), by restriction enzyme digestion with PmeI and BamHI and ligation using T4 DNA ligase.

Lentiviral particles were generated in HEK293T packaging cells by co-transfection of the target vector with the packaging psPAX2 (Addgene plasmid #12260) and envelope pMD2.G (Addgene #12259) vectors. Target cells were transduced as described (Perales-Clemente et al, 2008). Twenty-four hours after transduction, cells were selected for puromycin or hygromycin resistance.

Primary skin fibroblasts were immortalized by lentiviral transduction of pLOXTtag-iresTK (Addgene #12246).

All the utilized lentiviral vectors were a gift from Didier Trono.

## Quantitative PCR

For the quantification of mRNA levels, cDNA was retrotranscribed from total RNA extracted from mouse tissues or cultured cells (see above). Specific Gene Expression TaqMan assays (Invitrogen) were used for each of the transcripts of interest. Expression was calculated using the $\Delta\Delta C_t$ analysis using GAPDH as the reference.

## Native and denaturing electrophoresis and Western blot analysis

Samples for blue-native gel electrophoresis (BNGE) were prepared as described previously (Nijtmans et al, 2002; Wittig et al, 2006). Native samples were run through pre-cast NativePAGE 3–12% Bis–Tris gels, while Novex NuPAGE 4–12% Bis–Tris Gels (Life Technologies) were used for denaturing conditions.

For Western blot, samples were electroblotted to PVDF membranes and immunodetected using commercial specific antibodies from purchased from Abcam, Proteintech, Atlas Antibodies, Sigma, Santa Cruz, and Roche (Table 2). Polyclonal antibodies against APOPT1 were generated against the full-length recombinant human protein by Proteintech Group (Chicago, IL, USA; Catalog number: 27300-1-AP) and custom-made by Proteogenix (Schiltigheim, France).

Immunodetection signal intensities were quantified using ImageJ.

## Super-resolution fluorescence imaging and analysis

143B cells stably expressing APOPT1-GFP were transfected with different sub-compartment markers: mito-mScarlet (M), COX8A-DsRed (IM and TOMM20-DsRed (OM), or stained with MitoTracker CMXRos Red (Invitrogen) for IM + M, cells were then fixed with 3.7% formaldehyde. Coverslips were mounted with ProLong Diamond (Invitrogen).

Acquisition was performed using a N-SIM microscope system (Nikon) equipped with a SR Apo TIRF 100× 1.49 N.A. objective

**Table 1.  Oligonucleotides used for human APOPT1.**

| Name | Sequence |
|---|---|
| APOPT1 cDNA Fw | AATGCTGCCGTGCGCCGC |
| APOPT1 cDNA Rv | TGCTTCCTGTGGAAACCTGG |
| APOPT1-M1-PmeI-Fw | GTTTAAACC**ATG**CTGCCGTGCGCCGCG |
| APOPT1-M14-PmeI-Fw | GTTTAAACC**ATG**GTGGTCTTGCGGGCGG |
| APOPT1-201-HA-Rv | **TCA**AGCGTAATCTGGAACATCGTATGGGTAGTTGCTCCTCTTCTTTTGTTT**C** |
| APOPT1-203-HA-Rv | **TCA**AGCGTAATCTGGAACATCGTATGGGTAATGTTGCTTTCTGACCTTAC |
| APOPT1-201-GFP-pCR-NdeI-Rv | CATATGGTTGCTCCTCTTCTTTTGTTT**C** |
| APOPT1-203-GFP-pCR-NdeI-Rv | CATATGATGTTGCTTTCTGACCTTAC |

**Table 2.  Antibodies used in this study.**

| Antigen | Type | Dilution[a] | Company | Catalog number |
|---|---|---|---|---|
| ACO2 | Mouse monoclonal | 1:10,000 | Abcam | 6F12BD9 |
| APOPT1 | Rabbit polyclonal | 1:1,000 | Proteintech | 27300-1-AP |
| AIF | Mouse monoclonal | 1:1,000 | Santa Cruz | sc-13116 |
| AK2 | Rabbit monoclonal | 1:3,000 | Abcam | ab166901 |
| COX4 | Mouse monoclonal | 1:3,000 | Abcam | ab14744 |
| COX5A | Mouse monoclonal | 1:1,000 | Abcam | ab110262 |
| COX5B | Mouse monoclonal | 1:1,000 | Abcam | ab110263 |
| COX6B | Mouse monoclonal | 1:1,000 | Abcam | ab110266 |
| GAPDH | Mouse monoclonal | 1:5,000 | Abcam | ab8245 |
| GFP | Mouse monoclonal | 1:10,000 | Abcam | ab1218 |
| HA | Rat monoclonal | 1:1,000 | Roche | 11 867 431 001 |
| HSP70 | Mouse monoclonal | 1:1,000 | Abcam | ab2787 |
| MTCO1 | Mouse monoclonal | 1:3,000 | Abcam | ab14705 |
| MTCO2 | Mouse monoclonal | 1:10,000 | Abcam | ab110258 |
| MTCO2 | Rabbit polyclonal | 1:2,000 | Abcam | ab91317 |
| MTCO3 | Mouse monoclonal | 1:5,000 | Abcam | ab110259 |
| NDUFS1 | Rabbit polyclonal | 1:1,000 | Abcam | ab102552 |
| PNKD (MR-1S) | Rabbit polyclonal | 1:1,000 | Atlas Antibodies | HPA010134 |
| SCO2 | Rabbit polyclonal | 1:1,000 | Proteintech | 21223-1-AP |
| SDHB | Mouse monoclonal | 1:10,000 | Abcam | ab14714 |
| SOD2 | Mouse monoclonal | 1:2,000 | Abcam | ab16956 |
| BETA-TUBULIN | Mouse monoclonal | 1:10,000 | Sigma | T5201 |
| TOM20 | Rabbit monoclonal | 1:10,000 | Abcam | ab186734 |
| UBIQUITIN | Mouse monoclonal | 1:2,000 | Invitrogen | 13-1600 |
| UQCRC2 | Mouse monoclonal | 1:2,000 | Abcam | ab14745 |

[a]All primary antibodies were incubated overnight at 4°C.

and a DU897 Ixon camera (Andor Technologies). 3D-SIM image stacks were acquired with a Z-distance of 0.15 μm, and all the raw images were computationally reconstructed using the reconstruction slice system from NIS-Elements software (Nikon) keeping the same parameters. The co-localization analysis was performed using Imaris 9.0 XT software (Bitplane Scientific Software). For each image, threshold was applied in the same way using automated ROI threshold for the sub-compartments channel and adjusted in the same range for GFP staining. Pearson's coefficient in the co-localized volume was calculated for each image.

**Statistical analysis**

All numerical data are expressed as mean ± SEM (standard error). GraphPad PRISM software v.7.0e was used for the statistical analyses. To evaluate statistical significance of the differences between experimental groups, one-way ANOVA with Tukey's multiple comparisons or two-way ANOVA with Sidak's multiple comparison test was used. Unpaired, two-tailed Student's *t*-tests were used to evaluate statistical differences between two experimental conditions. *P* < 0.05 was considered statistically significant.

Exact *P*-values and number of animals or biological replicas (*n*) for each experiment are indicated in the figure legends. Power analysis was conducted assuming a significance level of 0.05 and a probability of detecting the effect of 80%. Animals were randomized to the different groups based on the appropriate genotype. No animals were excluded from analysis. No blinding procedure was used. We assumed normal distribution. No particular method was used to determine whether the data met assumptions of the statistical approach.

**Expanded View** for this article is available online.

## Acknowledgements

We are grateful to Dr. Lorenza Ronfani and co-workers from the Core Facility for Conditional Mutagenesis of the San Raffaele IRCCS (Milan, Italy). We thank Drs. Sara Vidoni, Marta Luna-Sanchez, Sukru Anil Dogan, and Aurora Gomez-Duran for their help and advice. Our work was supported by the Core Grant from the MRC (Grant MC_UU_00015/5); ERC Advanced under Grant FP7-322424 and NRJ-Institut de France Grant (to M.Z.); Telethon Foundation Italy Grant GGP15091 (to D.G.); Wellcome Trust Senior Clinical Research Fellowship 102770/Z/13/Z; and Lister Institute of Preventative Medicine RG87950 (to JAN).

## The paper explained

### Problem

APOPT1 is a protein mutated in young patients who accumulate severe brain lesions (cavitating leukoencephalopathy) associated with decreased amount and activity of cytochrome *c* oxidase (COX), the terminal component of the mitochondrial respiratory chain. However, in the past APOPT1 has been deemed as an apoptosis-inducing factor. To clarify these controversial points and shed light on the physiology of APOPT1, we created an APOPT1-less mouse and continued to investigate APOPT1 function and regulation in human cultured cells.

### Result

Phenotypical studies performed in the *Apopt1*$^{-/-}$ mice showed a neurological clinical phenotype. Biochemically, the knockout animals displayed significant isolated COX enzymatic and assembly deficiency in all the tested tissues. Expression of tagged forms of APOPT1 in patients' cells, when stable, restored the physiological amount and activity of COX. However, APOPT1 was found to be labile in cultured cells and mostly degraded as a precursor by the ubiquitin–proteasome system. However, under oxidative stress, the protein was stabilized and its mature form localized in the inner membrane of mitochondria, after cleavage of an N-terminal mitochondrial targeting sequence. In the absence of APOPT1, the mitochondrially encoded COX subunits showed a faster turnover which was exacerbated under oxidative stress. APOPT1 protected COX from oxidatively induced degradation.

### Impact

Taken together, the presented results demonstrate that APOPT1 participates in the formation of COX and protects the nascent enzyme by ROS-induced damage. Conversely, APOPT1 has no role in apoptosis. Based on the function and impact on human mitochondrial pathology, we propose to re-nominate APOPT1 as COX assembly factor 8, COA8. This is the first description of the regulation of COX assembly and function mediated by modulation of APOPT1 levels by the UPS, on the one hand, and by ROS, on the other hand. This could reflect the inability of adapting mitochondrial respiratory chain function to stress conditions in mitochondrial disease cases associated with loss-of-function mutations in APOPT1.

## Author contributions

AS performed most of the experiments on the biochemical, cell biology, and mouse phenotypic characterization. RC performed the histological and histochemical analysis. ASD and JAN performed the proteasome- and ubiquitin-related experiments. CB performed the super-resolution fluorescence microscopy experiments and quantifications. ECH and MPM provided MitoPQ and contributed to the corresponding experiments. DG, RC, and EB contributed the patient cell lines. CV supervised the creation and characterization of the mouse model. EF-V contributed to the molecular biology and biochemistry experiments. EF-V and MZ conceived the project, supervised the experimental workup, and wrote the manuscript. All the authors contributed to the final version of the paper.

## Conflict of interest

The authors declare that they have no conflict of interest.

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
