## [Review Process File · EMBO Molecular Medicine]

APOPT1/COA8 assists cox assembly and is oppositely regulated by UPS and ROS

Alba Signes, Raffaele Cerutti, Anna S. Dickson, Cristiane Benincá, Elizabeth C. Hinchy, Daniele Ghezzi, Rosalba Carrozzo, Enrico Bertini, Michael P. Murphy, James A. Nathan, Carlo Viscomi, Erika Fernandez-Vizarral and Massimo Zeviani

Review timeline:

Submission date:	20 July 2018
Editorial Decision:	17 September 2018
Revision received:	9 October 2018
Editorial Decision:	17 October 2018
Revision received:	15 November 2018
Editorial Decision:	19 November 2018
Revision received:	20 November 2018
Accepted:	21 November 2018

Editor: Céline Carret

Transaction Report:

1st Editorial Decision

17 September 2018

Thank you for the submission of your manuscript to EMBO Molecular Medicine. We have now heard back from the three referees whom we asked to evaluate your manuscript.

As you will see from the referees' reports pasted below, the study should in principle be published. However, both referees after cross-commenting feel that the mechanistic side of the data should be developed and ref. 2's suggestions should be followed. We also would like to encourage you to discuss the disease relevance and impact that these data have for patients to better fit with our scope.

Therefore, we would welcome the submission of a revised version within three months for further consideration and would like to encourage you to address all the criticisms raised as suggested to improve conclusiveness and clarity. Please note that EMBO Molecular Medicine strongly supports a single round of revision and that, as acceptance or rejection of the manuscript will depend on another round of review, your responses should be as complete as possible.

I look forward to receiving your revised manuscript.

***** Reviewer's comments *****

Referee #1 (Comments on Novelty/Model System for Author):

I did not mark "High" on novelty because the group published recently in AJHG that this gene has a role in COX assembly. However, this submission is much more mechanistic and I think it advances our knowledge in this relatively new protein (APOPT1/COA8). They used both patient cells and a KO mouse.

Referee #1 (Remarks for Author):

The manuscript by Signes and colleagues describe the characterization of APOPT1, which was previously shown by the same group to cause cavitating leukoencephalopathy associated with mitochondrial cytochrome c oxidase (COX) deficiency. In this study they generated an Apopt1 knockout mouse model, which showed decreased motor coordination, and endurance associated with reduced COX levels in several tissues. Expression of wild-type APOPT1 in control and patient derived cultured cells showed that this protein does not have a direct role in apoptosis and that this protein is necessary for proper COX assembly.

Overall the study is well performed and it adds to our knowledge of COX assembly. It shows clearly that COA8 has a role in COX assembly in vivo. I have relatively minor comments.

1. If they concluded that M14 is the endogenous initiation methionine, why use the M1 in the constructs?
2. In figure 4. Why is the stoichiometry between precursor/mature is so different when using HA or APOPT1.
3. I am not sure what is the physiological relevance of observing an over-expressed HA-tagged protein accumulating ubiquitinated forms when the proteasome is inhibited (Figure 6B). May be I am missing the point.
4. The following statement: "however, when the patient-derived fibroblasts were oxidatively challenged they showed a significant increase in ROS production compared to the controls (Melchionda et al., 2014), which argues in favor of a 'pro-oxidant state' in the absence of APOPT1." I am not sure I agree with the conclusion. The patient derived fibroblasts may have less ROS because of less COX. However, that led to an adaptation where less ROS scavenging mechanisms are required. These cells, when challenged, may be more sensitive to the oxidative stress.

Referee #2 (Remarks for Author):

In 2014, this group reported patient mutations in APOPT1 causing a COX deficiency. This study revealed a marked COX deficiency in muscle biopsy samples and in patient fibroblasts. The APOPT1-GFP fusion was reported to be mitochondria and suggested to be matrix localized based on the MTS cleavage. One highlight of the characterization of APOPT1 was that the protein was unstable in cells unless proteasomal degradation was blocked or the cells were treated with hydrogen peroxide. They concluded this study with the suggestion that APOPT1 exerts a protective role on COX during oxidative stress.

The present study builds on these 2014 observations and provides greater details in the various observations. They now generate an APOPT1 knockout mouse model, which shows the same COX deficiency associated with neurologic deficits. Thus, the APOPT1 KO mouse is an excellent model for neurologic testing studies. They demonstrate that APOPT1 depleted mouse tissues and human patient fibroblasts exhibit a diminution in COX subunits and MT-CO1 assembly intermediates. These studies are very well executed and the effects are marked. However, no major new insights are provided on the function of APOPT1. The investigators report again the oxidative accumulation of APOPT1 and show that APOPT1-deficient cells are oxidatively challenged with enhanced turnover of COX subunits. The significance of this present study is the more thorough

characterization of APOPT1-deficient cells, but many observations are reflected in their 2014 publication, which diminishes the novelty of the report.

The Cas9-mediated deletion of mouse APOPT1 is very well executed and discussed. Although they were unable to validate the KO by immunoblotting due to no protein detection, it is clear that a deletion was attained. The KO mice had reduced motor skills based on a myriad of solid studies. The COX biochemical studies on the KO tissues are well done, but it is surprising that no oxygen consumption measurements were done to address the consequences of reduced COX activity. They demonstrated that APOPT1-HA and GFP fusions expressed in HEK293 and 143B cells showed mitochondrial localization and loose association with the inner membrane. Using graded digitonin extractions followed by trypsin digestion, they conclude that the protein is IMS localized, but the data are not convincing and are not validated with other studies. The MTS of APOPT1 is cleaved in cells, so if APOPT1 is indeed IMS, it is one of the unusual IMS proteins with a cleaved MTS. Since vector borne-APOPT1 can rescue S2 or S6 skin fibroblast mutant cells, they could attempt to validate the IMS localization by testing whether replacing the APOPT1 MTS with a classical matrix MTS would give complementation. Is APOPT1 released by hypotonic solutions? It is important to validate the mitochondrial localization to better anticipate what functional role it may play in COX maturation under oxidative conditions. They postulate that COA8 exhibits a role in stabilizing the MT-CO2 module or the insertion of this module to the MT-CO1 module. A second way to validate an IMS localization is to assess interaction partners with affinity purified APOPT1. Does COA8 bind COX subunits or assembly factors such as SCO1, COA6, etc that affect the maturation of the MT-CO2 module? COA8 runs on a DDM BN gel with an apparent mass of ~40 kDa. Based on this apparent mass, they suggest that COA8 may interact with a protein of ~20 kDa. One cannot quantify the mass at the low end of BN gels, so this suggestion is not well based. However, a proteomic study of affinity purified COA8 might be very insightful and add impact to the current study. The BN study in Fig 4D revealed that under BN-PAGE conditions, COA8 does not associate with COX. However, BN conditions often dissociate weak interactions, so coIP studies are needed. The enhanced turnover of COX in APOPT1-deficient cells may arise from an oxidant-induced mechanism. One wonders whether OMA1 is activated in the deficient cells and contributes to the COX subunit turnover.

In summary, the description of the mouse model is excellent and worthy of publication. The apparent role of APOPT1 in stabilizing COX in oxidative conditions is striking and elucidating this mechanism would be a major contribution to the field. The impact of these studies would be markedly strengthened if some of the mentioned concerns were addressed.

A series of other concerns should be addressed.

1. The mature form of COA8 was stabilized by treatment with hydrogen peroxide. They conclude that this form must be mitochondrial, but many IMS proteins (e.g. COX19) are not stably localized in the IMS under stress conditions. They should validate that COA8 remains in the IMS under oxidant treatment. Under normal culture conditions, COA8 levels are labile suggesting proteolytic degradation. If COA8 is IMS localized, perhaps iAAA is responsible for the degradation or if matrix perhaps the LON protease mediates its diminution. Such studies could help validate the mitochondrial localization of COA8.
2. The APOPT1-deficient cells exhibit a down-regulation of COX subunits and subcomplexes. If COA8 is indeed IMS-localized, it could affect COX maturation through the association of COXVIb, which is oxidatively folded in the IMS. Is COXVIb association with COX impaired in mutant cells?
3. The mature human COA8 has four Cys residues, but it doesn't appear that any of them are totally conserved. They should comment on this. Are these Cys residues important for the candidate IMS localization of COA8?
4. References need editing for consistency in journal abbreviations and word spacing.

Referee #1 (Comments on Novelty/Model System for Author):

I did not mark "High" on novelty because the group published recently in AJHG that this gene has a role in COX assembly. However, this submission is much more mechanistic and I think it advances our knowledge in this relatively new protein (APOPT1/COA8). They used both patient cells and a KO mouse.

Referee #1 (Remarks for Author):

The manuscript by Signes and colleagues describe the characterization of APOPT1, which was previously shown by the same group to cause cavitating leukoencephalopathy associated with mitochondrial cytochrome c oxidase (COX) deficiency. In this study they generated an Apopt1 knockout mouse model, which showed decreased motor coordination and endurance associated with reduced COX levels in several tissues. Expression of wild-type APOPT1 in control and patient derived cultured cells showed that this protein does not have a direct role in apoptosis and that this protein is necessary for proper COX assembly.

Overall the study is well performed and it adds to our knowledge of COX assembly. It shows clearly that COA8 has a role in COX assembly in vivo. I have relatively minor comments.

We thank the Reviewer for the positive view about our work.

1. If they concluded that M14 is the endogenous initiation methionine, why use the M1 in the constructs?

We did not see any changes in expression levels or protein size using the constructs starting from 'M1' or 'M14'. However, we reasoned that the sequence in between those two positions is the APOPT1 mRNA 5'-UTR so that by keeping it we would maintain a more physiological structure and possibly expression of the protein.

2. In figure 4. Why is the stoichiometry between precursor/mature is so different when using HA or APOPT1.

The samples ran in lanes 2 and 3 (labelled as 'S6' and 'S2') in the Western blot image shown in figure 4A are from the patients described in Melchionda et al. 2014. The mutations predict truncated APOPT1 protein products, and, therefore the absence of the full-length precursor protein. However, the anti-APOPT1 antibody recognizes in the patient samples a protein band of the same size as the APOPT1 precursor. Obviously, the signal detected in the patients' samples cannot correspond to the APOPT1 precursor but to another protein. The band in the wild type may well correspond to both the Apopt1 precursor and to the unknown protein shared with the patients samples. Conversely, the anti-HA antibody detects only the tagged precursor and mature species with no "noise" derived from cross-reacting contaminants.

3. I am not sure what is the physiological relevance of observing an over-expressed HA-tagged protein accumulating ubiquitinated forms when the proteasome is inhibited (Figure 6B). May be I am missing the point.

The experiment demonstrates that HA tagged APOPT1 is functional and the accumulation and ubiquitination of the APOPT1-HA precursor following proteasome inhibition shows that this protein is very tightly regulated within the cytosol, which is in contrast to the effect of H2O2 where the cleaved mitochondrial form accumulates. These experiments also show that APOPT1 stabilisation following proteasome inhibition is a direct effect on the protein (as it is ubiquitinated), rather than an indirect effect of pharmacological treatment.

4. The following statement: "however, when the patient-derived fibroblasts were oxidatively challenged they showed a significant increase in ROS production compared to the controls (Melchionda et al., 2014), which argues in favor of a 'pro-oxidant state' in the absence of APOPT1." I am not sure I agree with the conclusion. The patient derived fibroblasts may have less ROS

because of less COX. However, that led to an adaptation where less ROS scavenging mechanisms are required. These cells, when challenged, may be more sensitive to the oxidative stress.

This could indeed be a valid hypothesis. However, the basal H₂O₂ production in patient-derived fibroblasts was not lower than in control fibroblasts (Melchionda et al., 2014). Also, we could not detect differences in basal ROS production in isolated mitochondria from heart and brain of KO and WT mice. In addition, we tested the steady-state levels of SOD2 as well as those of ACO2 (mitochondrial aconitase, which serves as a marker of oxidative stress) by Western blot in mouse liver, brain and skeletal muscle and we did not find differences between Apopt1^{wt} (wt = +/+ and +/-) and Apopt1^{-/-} (see image below showing representative blots obtained with female mice samples and the result of the densitometric quantification summarized in the graph).

To further address the point raised by the Reviewer, we have analysed the steady-state levels of SOD2 and aconitase in the patient-derived immortalized fibroblasts, the same cell lines complemented with wild-type APOPT1, and in two different controls. As it can be seen below, the levels of SOD2 were not lower in the patient-derived fibroblasts and neither SOD2 nor ACO2 showed significant differences in abundance in the three types of analysed cell lines (APOPT1-null, COMPLEMENTED and CONTROL fibroblasts).

All these observations suggest that there is no decrease of ROS in basal states and no decrease in antioxidant stress defences in APOPT1-less cells and tissues.

Referee #2 (Remarks for Author):

In 2014, this group reported patient mutations in APOPT1 causing a COX deficiency. This study revealed a marked COX deficiency in muscle biopsy samples and in patient fibroblasts. The APOPT1-GFP fusion was reported to be mitochondria and suggested to be matrix localized based on the MTS cleavage. One highlight of the characterization of APOPT1 was that the protein was unstable in cells unless proteasomal degradation was blocked or the cells were treated with hydrogen peroxide. They concluded this study with the suggestion that APOPT1 exerts a protective role on COX during oxidative stress.

The present study builds on these 2014 observations and provides greater details in the various

observations. They now generate an APOPT1 knockout mouse model, which shows the same COX deficiency associated with neurologic deficits. Thus, the APOPT1 KO mouse is an excellent model for neurologic testing studies. They demonstrate that APOPT1 depleted mouse tissues and human patient fibroblasts exhibit a diminution in COX subunits and MT-CO1 assembly intermediates. These studies are very well executed and the effects are marked. However, no major new insights are provided on the function of APOPT1. The investigators report again the oxidative accumulation of APOPT1 and show that APOPT1-deficient cells are oxidatively challenged with enhanced turnover of COX subunits. The significance of this present study is the more thorough characterization of APOPT1-deficient cells, but many observations are reflected in their 2014 publication, which diminishes the novelty of the report.

The Cas9-mediated deletion of mouse APOPT1 is very well executed and discussed. Although they were unable to validate the KO by immunoblotting due to no protein detection, it is clear that a deletion was attained. The KO mice had reduced motor skills based on a myriad of solid studies. The COX biochemical studies on the KO tissues are well done, but it is surprising that no oxygen consumption measurements were done to address the consequences of reduced COX activity.

We did some measurements from knockout mouse heart and liver isolated mitochondria. We observed a reduction of 33% in oxygen consumption induced by succinate, compared with the wild type, which is consistent with the COX deficiency displayed by the mouse tissues. We did not perform a systematic set of measurements as we considered that the kinetic and histochemical analyses were sufficient to demonstrate the significant isolated COX deficiency associated with the absences of COA8.

They demonstrated that APOPT1-HA and GFP fusions expressed in HEK293 and 143B cells showed mitochondrial localization and loose association with the inner membrane. Using graded digitonin extractions followed by trypsin digestion, they conclude that the protein is IMS localized, but the data are not convincing and are not validated with other studies.

We think that there is some terminological confusion about the localization of APOPT1. In the experiments shown in figure 4 we demonstrate that APOPT1 is associated with the inner mitochondrial membrane, after cleavage of the MTS. This conclusion is provided by the evidence that the protein is present in the membrane fraction in basal conditions, and is partially extracted only by treatment with 0.1M Na₂CO₃, pH 10.5. In addition, the experiments based on the sensitivity of different proteins to trypsin proteolysis in the presence of increasing concentrations of the mild detergent digitonin, show that APOPT1 is partially sensitive to trypsin+digitonin, in a way very similar if not identical to AIF (apoptosis inducing factor), which is a protein bound to the inner membrane of mitochondria but protruding towards the intermembrane space. Other inner membrane-bound proteins, such as CO1, which is embedded in the membrane, or matrix proteins (Aco2) are instead completely protected by trypsin, unless the membranes are solubilized by Triton XI00. This behaviour to trypsin digestion is indeed similar to a known IMS soluble protein (AK2) which however does not co-fraction with the membranes. Therefore, the combination of these two results, co-fraction with membranes and partial sensitivity to trypsin, indicates that, like AIF, APOPT1 is indeed an inner membrane protein facing the IMS. This concept has now been clearly explained in the manuscript.

The MTS of APOPT1 is cleaved in cells, so if APOPT1 is indeed IMS, it is one of the unusual IMS proteins with a cleaved MTS. Since vector borne-APOPT1 can rescue S2 or S6 skin fibroblast mutant cells, they could attempt to validate the IMS localization by testing whether replacing the APOPT1 MTS with a classical matrix MTS would give complementation.

We would like to draw the attention of Reviewer 2 to the fact that we never asserted that APOPT1 is a soluble IMS protein. Of course, we do not know whether the protein is completely translocated through the inner membrane, cleaved in the matrix and repositioned in the inner membrane, or simply inserted in the inner membrane where the N-terminal MTS is then cleaved off. Possibly, the latter is the most likely mechanism, known to act also for other proteins, for instance PINK1, MICU1 and possibly AIF itself.

Is APOPT1 released by hypotonic solutions? It is important to validate the mitochondrial localization to better anticipate what functional role it may play in COX maturation under oxidative conditions.

We would like to emphasize that we find APOPT1 only in the membrane fraction of isolated mitochondria even after the samples are vigorously sonicated, indicating a tight physical binding to the membrane that only strong alkalization and increase of the ionic strength by carbonate can partially disrupt.

They postulate that COA8 exhibits a role in stabilizing the MT-CO2 module or the insertion of this module to the MT-CO1 module. A second way to validate an IMS localization is to assess interaction partners with affinity purified APOPT1. Does COA8 bind COX subunits or assembly factors such as SCO1, COA6, etc that affect the maturation of the MT-CO2 module?

We indeed performed immunopurification experiments of APOPT1^{HA} and APOPT1^{GFP} and tested the purified fractions for the presence of COX structural subunits, but failed to detect any of them.

COA8 runs on a DDM BN gel with an apparent mass of ~40 kDa. Based on this apparent mass, they suggest that COA8 may interact with a protein of ~20 kDa. One cannot quantify the mass at the low end of BN gels, so this suggestion is not well based.

We thank the Reviewer for this remark and agree that this estimation of the molecular weight is not accurate enough. We have eliminated this statement from the text and substituted it with the following sentence in page 12: "This suggests that both versions of APOPT1 are not stably interacting in a high-molecular weight complex, including the COX assembly intermediates containing MT-CO1 or MT-CO2."

However, a proteomic study of affinity purified COA8 might be very insightful and add impact to the current study. The BN study in Fig 4D revealed that under BN-PAGE conditions, COA8 does not associate with COX. However, BN conditions often dissociate weak interactions, so coIP studies are needed.

In addition to the unlabelled co-IP experiments with the targeted detection of COX structural subunits, we also performed SILAC experiments combined with immunopurification of APOPT1^{GFP}. Again we could not find any significant interaction with mitochondrial proteins, suggesting that APOPT1 does not establish permanent or long-lasting stable interactions with partner proteins. This is also suggested by the fast migration of APOPT1 in BNGE with DDM and digitonin, which again indicates lack of stable interactions. It is possible that more subtle interactions can be detected by different techniques (e.g. cross-linking, bio-ID), which will require a dedicated optimization.

The enhanced turnover of COX in APOPT1-deficient cells may arise from an oxidant-induced mechanism. One wonders whether OMA1 is activated in the deficient cells and contributes to the COX subunit turnover.

In summary, the description of the mouse model is excellent and worthy of publication. The apparent role of APOPT1 in stabilizing COX in oxidative conditions is striking and elucidating this mechanism would be a major contribution to the field. The impact of these studies would be markedly strengthened if some of the mentioned concerns were addressed.

A series of other concerns should be addressed.

1. The mature form of COA8 was stabilized by treatment with hydrogen peroxide. They conclude that this form must be mitochondrial, but many IMS proteins (e.g. COX19) are not stably localized in the IMS under stress conditions. They should validate that COA8 remains in the IMS under oxidant treatment.

As we tried to explain above, we strongly believe that APOPT1 is bound to the inner membrane of mitochondria rather than free in the IMS. However, we performed an experiment based on immunofluorescence of APOPT1-GFP in cells under H₂O₂ treatment and did not observe any dispersion of the signal or change of the "mitochondrial pattern" of the APOPT1-specific IF.

Under normal culture conditions, COA8 levels are labile suggesting proteolytic degradation. If COA8 is IMS localized, perhaps iAAA is responsible for the degradation or if matrix perhaps the LON protease mediates its diminution. Such studies could help validate the mitochondrial localization of COA8.

This is clearly an interesting question but we think it deserves a dedicated project. AIF, for instance, is cleaved by calpain to produce the soluble form that eventually triggers apoptosis. We have clearly shown that APOPT1 is mainly degraded as a precursor in the cytosol. At this stage of the study, the further intramitochondrial degradation of the protein is clearly interesting but perhaps not prioritary.

2. The APOPT1-deficient cells exhibit a down-regulation of COX subunits and subcomplexes. If COA8 is indeed IMS-localized, it could affect COX maturation through the association of COXVIb, which is oxidatively folded in the IMS. Is COXVIb association with COX impaired in mutant cells?

As shown in Figure 3D, the steady-state levels of COX6B1 are very low in the knockout mouse liver and skeletal muscle. However, they are as reduced as other COX subunits that are assembled late (MT-CO3) or in the intermediate steps of COX assembly (MT-CO2 and COX5B). Therefore, the association of the subunit is impaired, but it does not seem to be a specific phenomenon for COX6B1.

3. The mature human COA8 has four Cys residues, but it doesn't appear that any of them are totally conserved. They should comment on this. Are these Cys residues important for the candidate IMS localization of COA8?

Thank you for this comment. The "oxidized cys" hypothesis has to be tested for instance by mutating the cys residues, again an experiment that is in the pipeline of future investigation on this intriguing protein. However, we notice that even in the human sequence the distribution of the cys residues does not follow the consensus of IMS proteins (Cx3C or Cx9C).

4. References need editing for consistency in journal abbreviations and word spacing.
OK, thank you.

2nd Editorial Decision

17 October 2018

Thank you for the submission of your revised manuscript to EMBO Molecular Medicine. We have now received the enclosed reports from the referees that were asked to re-assess it. As you will see the reviewers are now globally supportive and I am pleased to inform you that we will be able to accept your manuscript pending minor editorial amendments and a response to referee 2. We would like to encourage you to add experimental evidence as suggested during the 1st review. Failing this, the conclusions have to be softened. Please make sure to discuss the referee comments in a point-by-point letter.

I look forward to reading a new revised version of your manuscript as soon as possible.

***** Reviewer's comments *****

Referee #1 (Comments on Novelty/Model System for Author):

It is an interesting and well-performed study of a disease-related cytochrome c oxidase assembly factor with mechanistic insights.

Referee #1 (Remarks for Author):

The authors have addressed my concerns

Referee #2 (Comments on Novelty/Model System for Author):

The impact is somewhat limited based on their previous report. The mouse model is excellent and it validated their earlier work. Beyond that, the impact of the work is only "medium"

Referee #2 (Remarks for Author):

This reviewer is disappointed that the investigators ignored all requests to substantiate their studies to gain rigor. Certain conclusions are based on weak data such as the protrusion of Coa8 into the IMS. In the revision, their statement "Taken together, these observations clearly indicate that APOPT1 is a protein tightly associated with the inner membrane, which is protruding into the IMS." is unacceptable without qualification since they did not validate the one observation. Without additional studies to substantiate observations, this revision is only partially above the bar for acceptance. They must soften their conclusion that Coa8 protrudes into the IMS. This is such an important point in considering how Coa8 may exert its stabilizing function.

2nd Revision - authors' response

15 November 2018

Referee #2 (Comments on Novelty/Model System for Author):

The impact is somewhat limited based on their previous report. The mouse model is excellent and it validated their earlier work. Beyond that, the impact of the work is only "medium"

Referee #2 (Remarks for Author):

This reviewer is disappointed that the investigators ignored all requests to substantiate their studies to gain rigor. Certain conclusions are based on weak data such as the protrusion of Coa8 into the IMS. In the revision, their statement "Taken together, these observations clearly indicate that APOPT1 is a protein tightly associated with the inner membrane, which is protruding into the IMS." is unacceptable without qualification since they did not validate the one observation. Without additional studies to substantiate observations, this revision is only partially above the bar for acceptance. They must soften their conclusion that Coa8 protrudes into the IMS. This is such an important point in considering how Coa8 may exert its stabilizing function.

Our apologies for not addressing the question of the localization in a more rigorous way before. We thank Referee #2 for insisting in clarifying this point as it made us aware of the important functional implications related to APOPT1 topology within mitochondria. We have now re-addressed the question of APOPT1 sub-mitochondrial localization by performing an additional biochemical experiment of mitochondrial extraction, subfractionation by hypotonic shock and protease protection assays in APOPT1^{HA} cells. In addition, we have performed super-resolution fluorescence imaging, without using antibodies, in APOPT1^{GFP} cells (showed in the new Figure 4D). After treating the mitochondria with the hypotonic buffer, and replying to one of the Referee's original questions, we did not observe any release of APOPT1 to the supernatants while the soluble intermembrane space protein AK2 was clearly detectable (new Figure 4C). Also, a small peptide derived from SCO2 after trypsin digestion was released to the supernatant. The fluorescence-based imaging confirmed that APOPT1-GFP is contained within the matrix compartment and does not protrude beyond the contour of the inner membrane.

After evaluating these new data, we now conclude that the APOPT1 C-terminus is in the mitochondrial matrix. This part of the results has been re-written in pages 11 and 12 and the new results are shown in the new Figure 4.

Again, we thank Referee #2 for prompting us to obtain additional important information on this issue.

Corresponding Author Name: Massimo Zeviani and Erika Fernandez-Vizarra

Manuscript Number: EMM-2018-09582